# Chromosomal instability by mutations in the novel minor spliceosome component *CENATAC*

Bas de Wolf[1], Ali Oghabian[2,†,§], Maureen V Akinyi[2,†] iD, Sandra Hanks[3,†], Eelco C Tromer[1,4,¶],
Jolien J E van Hooff[1,4,††], Lisa van Voorthuijsen[5], Laura E van Rooijen[4], Jens Verbeeren[2],
Esther C H Uijttewaal[1], Marijke P A Baltissen[5], Shawn Yost[3], Philippe Piloquet[6], Michiel Vermeulen[5] iD,
Berend Snel[4], Bertrand Isidor[6], Nazneen Rahman[3,‡], Mikko J Frilander[2,*,‡] iD & Geert J P L Kops[1,**,‡] iD

## Abstract

**Aneuploidy is the leading cause of miscarriage and congenital birth defects, and a hallmark of cancer. Despite this strong association with human disease, the genetic causes of aneuploidy remain largely unknown. Through exome sequencing of patients with constitutional mosaic aneuploidy, we identified biallelic truncating mutations in *CENATAC* (*CCDC84*). We show that CENATAC is a novel component of the minor (U12-dependent) spliceosome that promotes splicing of a specific, rare minor intron subtype. This subtype is characterized by AT-AN splice sites and relatively high basal levels of intron retention. CENATAC depletion or expression of disease mutants resulted in excessive retention of AT-AN minor introns in ∼ 100 genes enriched for nucleocytoplasmic transport and cell cycle regulators, and caused chromosome segregation errors. Our findings reveal selectivity in minor intron splicing and suggest a link between minor spliceosome defects and constitutional aneuploidy in humans.**

**Keywords** aneuploidy; CCDC84; CENATAC; minor spliceosome
**Subject Categories** Cell Cycle; Genetics, Gene Therapy & Genetic Disease; RNA Biology
**The EMBO Journal (2021) 40: e106536**

## Introduction

Chromosome segregation errors in mitosis or meiosis lead to aneuploidy, a karyotype that deviates from an exact multiple of the haploid set of chromosomes. Aneuploidy is the leading cause of congenital birth defects and associated with ∼ 35% of all spontaneous human abortions (Nagaoka *et al*, 2012). Furthermore, roughly 70% of human tumors are aneuploid, making it one of the most common genomic alterations in cancer (Duijf & Benezra, 2013; Knouse *et al*, 2017). Despite this common association of aneuploidy with human disease, little is known about its genetic causes. The study of aneuploidy-associated hereditary disorders can be instrumental in uncovering these causes.

Mosaic variegated aneuploidy (MVA; OMIM: 257300) is a rare autosomal recessive disorder characterized by mosaic aneuploidies in multiple tissues. Patients often present with microcephaly, developmental delay, various congenital abnormalities, and childhood cancers (García-Castillo *et al*, 2008). Pathogenic mutations in *BUB1B*, *CEP57*, or *TRIP13*, have been identified in roughly half of all MVA patients (Hanks *et al*, 2004; Matsuura *et al*, 2006; Snape *et al*, 2011; Yost *et al*, 2017). These genes have well-documented roles in chromosome segregation (Suijkerbuijk *et al*, 2010; Sacristan & Kops, 2015; Vader, 2015; Zhou *et al*, 2016). All three gene products (BUBR1, CEP57, and TRIP13) promote spindle assembly checkpoint (SAC) function (Wang *et al*, 2014; Musacchio, 2015; Ma *et al*, 2016; Zhou *et al*, 2016; Alfieri *et al*, 2018), and BUBR1 and CEP57 additionally ensure correct kinetochore–microtubule attachment (Emanuele & Stukenberg, 2007; Sacristan & Kops, 2015). As predicted, such mitotic processes are defective in cells from MVA

---

1 Oncode Institute, Hubrecht Institute - Royal Academy of Arts and Sciences and University Medical Centre Utrecht, Utrecht, The Netherlands
2 Institute of Biotechnology, Helsinki Institute of Life Science, University of Helsinki, Helsinki, Finland
3 Division of Genetics and Epidemiology, Institute of Cancer Research, London, UK
4 Theoretical Biology and Bioinformatics, Biology, Science Faculty, Utrecht University, Utrecht, The Netherlands
5 Oncode Institute, Department of Molecular Biology, Faculty of Science, Radboud Institute for Molecular Life Science, Radboud University Nijmegen, Nijmegen, The Netherlands
6 Service de Génétique Médicale, Unité de génétique Clinique, CHU Hotel Dieu, Nantes Cedex, France
  *Corresponding author. Tel: +358 2941 59509; E-mail: mikko.frilander@helsinki.fi
  **Corresponding author. Tel: +313 0212 1907; E-mail: g.kops@hubrecht.eu
  †These authors contributed equally to this work
  ‡These authors contributed equally to this work as senior authors
  §Present address: Faculty of Medicine, Research Programs Unit, University of Helsinki, Helsinki, Finland
  ¶Present address: Department of Biochemistry, University of Cambridge, Cambridge, UK
  ††Present address: Unité d'Ecologie, Systématique et Evolution, CNRS, Université Paris-Sud, Université Paris-Saclay, AgroParisTech, Orsay, France

---

patients carrying biallelic mutations in these genes, explaining the chromosomal instability (CIN) phenotype and resulting aneuploid karyotypes. CIN can also result from mutations in regulators of expression of mitotic genes. For example, mutations in the retinoblastoma gene (*RB1*) cause CIN by overexpression of the SAC protein MAD2 (Hernando *et al*, 2004; Sotillo *et al*, 2007; Schvartzman *et al*, 2011). In this work, we show that chromosome segregation errors can be caused by a specific defect in minor intron splicing, another process governing correct gene expression.

While the conventional, major spliceosome targets most (> 99.5%) human introns, the minor spliceosome recognizes and excises only a small subset (~ 700 introns) (Turunen *et al*, 2013a; Moyer *et al*, 2020). These minor introns (also called U12-type introns) have highly conserved 5′ splice site (5′ss) and branch point (BPS) sequences that are longer and differ at the sequence level from the respective sequences in major (U2-type) introns. Most minor introns have AT-AC or GT-AG terminal dinucleotides (24 and 69%, respectively) (Sheth *et al*, 2006; Moyer *et al*, 2020). In addition, the 3′ terminal nucleotide can vary, thus giving rise to AT-AN and GT-AN classes of minor introns (Levine & Durbin, 2001; Dietrich *et al*, 2005). For simplicity, we refer to these as A- and G-type introns, respectively. Thus far, there has been no indication of mechanistic or functional differences between the minor intron subtypes.

Minor intron "host" genes, the position of the minor intron within the gene, and intron subtypes, are all evolutionarily conserved (Burge *et al*, 1998; Abril *et al*, 2005; Sheth *et al*, 2006; Alioto, 2007; Moyer *et al*, 2020). Despite this high conservation, the functional significance of minor introns has remained elusive. Elevated levels of unspliced minor introns in various cell types have been reported, giving rise to the hypothesis that these are rate-limiting controls for the expression of their host genes (Patel *et al*, 2002; Younis *et al*, 2013; Niemelä & Frilander, 2014; Niemelä *et al*, 2014). Nevertheless, the overall significance of the elevated intron retention (IR) levels has been questioned particularly at individual gene level (Singh & Padgett, 2009).

The overall architecture of the minor and major spliceosomes is highly similar. Both are composed of five small ribonucleoprotein (snRNP) complexes containing small nuclear RNA (snRNA) molecules and a large number of protein components. One of the snRNAs (U5) is shared between the spliceosomes, while U1, U2, U4, and U6 snRNAs are specific to the major spliceosome, and U11, U12, U4atac, and U6atac snRNAs to the minor spliceosome. Introns are initially recognized by the U1 and U2 snRNPs (major spliceosome) or by the U11/U12 di-snRNP (minor spliceosome), followed by the entry of the U4/U6.U5 or U4atac/U6atac.U5 tri-snRNP and subsequent architectural changes leading to catalytic activation of the spliceosome (Turunen *et al*, 2013a). At the protein level, the main difference between the spliceosomes is in the composition of the U11/U12 di-snRNP that contains seven unique protein components that are needed for recognition of the unique minor intron splice sequences (Will *et al*, 2004). In contrast, the protein composition of the minor and major tri-snRNPs appears similar, but rigorous comparative analyses have been difficult due to the ~ 100-fold lower cellular abundance of the minor tri-snRNP (Schneider *et al*, 2002).

Here, we report that germline mutations in a novel component of the minor spliceosome (*CENATAC/CCDC84*) cause chromosomal instability in MVA patients. We identify CENATAC as a minor spliceosome-specific tri-snRNP subunit that promotes the splicing of A-type minor introns, but hardly contributes to G-type minor intron splicing. We show that CENATAC depletion or disease mutations result in increased A-type minor IR and mitotic chromosome congression defects. Congression defects are also seen when another minor spliceosome component is depleted, suggesting that the chromosome segregation errors and aneuploidy observed in MVA patient cells are secondary effects of defective minor intron splicing.

# Results

## Biallelic truncating mutations in *CENATAC* (*CCDC84*) cause MVA

To search for additional causes of MVA, we performed exome sequencing and variant analyses on MVA patients and family members, as previously described (Yost *et al*, 2017). We identified biallelic truncating mutations in coiled-coil domain-containing 84 (*CCDC84*, hereafter named *CENATAC*, for centrosomal AT-AC splicing factor, see below) in two affected siblings with 7.3 and 8.5% aneuploid blood cells, respectively (Figs 1A and EV1). Both siblings were alive at 47 and 33 years of age and had microcephaly, mild developmental delay, and mild maculopathy. Neither individual had short stature, dysmorphism, or cancer. Each parent was heterozygous for one of the mutations, and the unaffected sibling had neither mutation. Moreover, the mutations were absent from the ExAC and ICR1000 series and we estimated the chance of an individual having two truncating *CENATAC* mutations to be $4.8 \times 10^{-10}$ (Fitzgerald *et al*, 2015). We therefore consider it very likely that the *CENATAC* mutations are the cause of the siblings' phenotype. The paternal and maternal mutations (mutation 1 and mutation 2, respectively) both result in the creation of novel splice sites that lead to a frame-shift and the loss of the C-terminal 64 amino acids of CENATAC (Fig 1B and Appendix Fig S1). Although expression of the mutant alleles was very low in the parental cells, expression of the maternal allele was elevated in the cells of patient 1 (hereafter called patient) and was responsible for the low expression of wild-type protein in these cells due to infrequent recognition of the original splice site (Fig 1C and Appendix Fig S1C).

*CENATAC* is an essential gene whose product has previously been reported to interact with pre-mRNA splicing factors and to localize to centrosomes where it suppresses centriole over-duplication and spindle multipolarity (Hart *et al*, 2015; Wang *et al*, 2019). Analysis of *CENATAC* sequence conservation in metazoan species revealed the presence of two N-terminal C2H2 zinc fingers and four well-conserved C-terminal sequence motifs, of which the two most C-terminal ones are lost as a result of the patient mutations (Fig 1B and Appendix Fig S2).

## CENATAC promotes error-free chromosome segregation

Live imaging of chromosome segregation in *CENATAC* mutant patient lymphoblasts stably expressing H2B-mNeon (Yost *et al*, 2017) revealed a mild chromosomal instability phenotype, consistent with the modest levels of aneuploidy in blood cells of these patients (Figs 1A and 2A). To examine whether *CENATAC* patient mutations cause chromosomal instability, we expressed mutant *CENATAC* alleles in HeLa cells in which the endogenous loci were modified to express AID-degron-tagged CENATAC (HeLa^*EGFP-AID-CENATAC*,

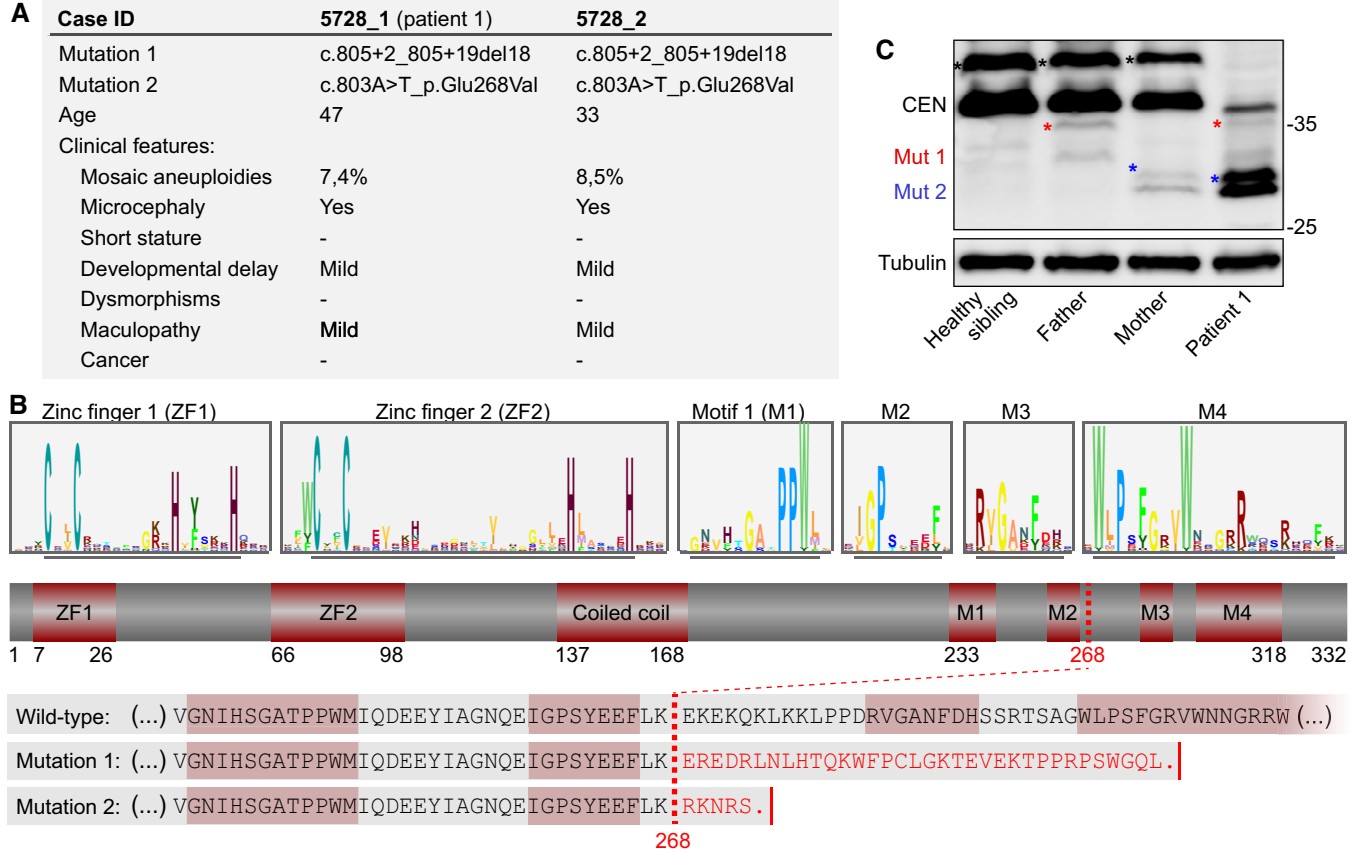

**Figure 1. Biallelic truncating mutations in *CENATAC* (*CCDC84*) cause MVA.**

A  Clinical phenotypes of *CENATAC* (*CCDC84*) mutant patients. See also Fig EV1.

B  Schematic representation of CENATAC annotated with zinc fingers (ZF1 and ZF2), predicted coiled-coil, and conserved motifs 1–4 (M1–M4). Upper: sequence logos of both zinc fingers and the conserved residues defining motifs 1–4 (underlined). See Appendix Fig S2 for the full-length logo. Lower: C-terminal protein sequences of wild-type and MVA mutant CENATAC. The MVA truncation site is indicated by the red dotted line; the four conserved motifs are outlined in red.

C  CENATAC and tubulin immunoblots of lysates from lymphoblasts of patient 1 and relatives. Wild-type and truncated, mutant proteins are indicated. Wild-type CENATAC (CEN): 38 kDa, Mut1: 34.5 kDa (father), Mut2: 31.1 kDa (mother). Phosphorylated CENATAC is indicated with asterisks: black (wild-type), red (Mut1), or blue (Mut2) (Wang *et al*, 2019).

Appendix Fig S3) (Nishimura *et al*, 2009). Efficient depletion of CENATAC through a combination of siRNA treatment and auxin addition caused chromosome congression defects and a subsequent mitotic arrest (Fig 2B and C, and Appendix Fig S3). This phenotype was fully rescued upon re-expression of wild-type but not MVA mutant CENATAC (Fig 2C and Appendix Figs S4 and S5), indicating that both MVA mutants are defective for CENATAC's function in mitotic chromosome congression. MVA mutant CENATAC caused a similar mitotic phenotype when expressed in near-diploid DLD-1 cells (Appendix Fig S6). CENATAC alleles missing either of the two most C-terminal conserved motifs that are absent from MVA mutant CENATAC (Fig 1B, motifs 3 and 4) did not rescue the mitotic defects. Instead, the expression of the MVA or motif 3/4 mutants exacerbated the phenotype, suggesting that these proteins dominantly repressed the function of any residual wild-type protein (Fig 2C). Mutations in the zinc fingers or deletion of motifs 1 or 2 only partly compromised CENATAC function (Figs 1B and 2C).

Live imaging of HeLa^EGFP-AID-CENATAC cells with fluorescently labeled chromatin and microtubules revealed that the chromosome congression defect upon CENATAC depletion preceded the previously described loss of spindle bipolarity (Figs 2D and E, and EV2A and B, Movies EV1 and EV2) (Wang *et al*, 2019). In addition, we did not observe centriole over-duplication in CENATAC-depleted cells (Fig EV2C and D). This is in contrast to what was recently reported for *CENATAC* knockout cells (Wang *et al*, 2019), raising the possibility that centriole over-duplication is a cumulative effect of prolonged CENATAC loss. Our attempts to examine this failed, as we were unable to create *CENATAC* knockout cells, consistent with it being an essential human gene (Blomen *et al*, 2015; Hart *et al*, 2015; Wang *et al*, 2015). Taken together, these data show that CENATAC directly or indirectly promotes chromosome congression in mitosis (in a manner likely unrelated to its role in maintaining spindle bipolarity) and that MVA mutant CENATAC is a defective variant.

**CENATAC is a novel component of the minor spliceosome**

To investigate in which processes CENATAC plays a role, we performed a genome-wide, evolutionary co-occurrence analysis.

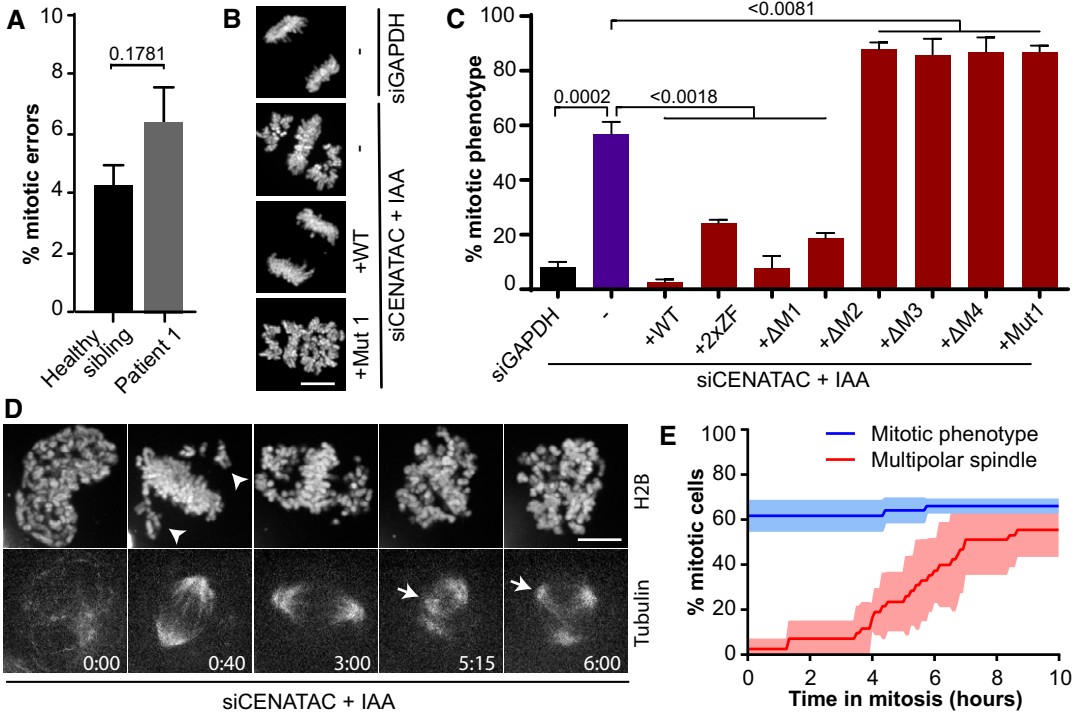

Figure 2. CENATAC promotes error-free chromosome segregation.

A Quantification of chromosome segregation errors of patient and control lymphoblasts expressing H2B-mNeon (four biological replicates, > 200 cells in total per condition).

B Representative images of H2B-mNeon-expressing HeLa$^{EGFP-AID-CENATAC}$ cells depleted of GAPDH (upper) or CENATAC (middle and lower) with or without re-expression of CENATAC variants as indicated. IAA, 3-indoleacetic acid. Scale bar, 10 μm.

C Quantification of mitotic defects as in (B) of H2B-mNeon-expressing HeLa$^{EGFP-AID-CENATAC}$ cells treated as indicated (three or five biological replicates, > 85 cells in total per condition). For 2xZF, the four zinc-finger cysteines were mutated to alanines; for Δ1–4, the corresponding motif was removed.

D Representative stills of HeLa$^{EGFP-AID-CENATAC}$ cells expressing H2B-mNeon and depleted of CENATAC. Microtubules were visualized with SiR-Tubulin. Arrowheads and arrows indicate non-congressed chromosomes and supernumerary spindle poles, respectively. Scale bar, 5 μm. Time in hours. See Fig EV2 for the control condition. See also Movies EV1 and EV2. IAA, 3-indoleacetic acid.

E Quantification of the mitotic phenotype and multipolar spindle formation in time in H2B-mNeon-expressing HeLa$^{EGFP-AID-CENATAC}$ cells treated as in (D) (three biological replicates, > 44 cells in total per condition). See also Fig EV2.

Data information: In (A, C, E), data are presented as mean ± SEM. P-values were calculated with unpaired Student's t-tests.

Genes that function in the same biochemical process experience similar evolutionary pressures and therefore tend to co-evolve, i.e., they are lost or retained in a coherent fashion (Pellegrini *et al*, 1999). Genomes from a set of 90 informative eukaryotic species (Table EV1) were mined for the presence or absence of *CENATAC* orthologs (Hooff *et al*, 2017). This provided a phylogenetic absence/presence profile that was used in an unbiased genome-wide query for genes with similar phylogenetic profiles (Fig 3A). The resulting list of genes most strongly co-occurring with *CENATAC* was significantly enriched for components of the minor (U12-dependent) spliceosome complex, including the recently discovered SCNM1 (Bai *et al*, 2021) (Fig 3B, Table EV2). We thus reasoned that CENATAC may play a role in splicing by the minor spliceosome.

As predicted by our co-evolution analysis and in agreement with a previous high-throughput screen (Hart *et al*, 2015; Hein *et al*, 2015; Huttlin *et al*, 2015, 2017), mass spectrometry analysis of proteins co-purifying with CENATAC identified several known spliceosome components that are shared by both the major and minor spliceosomes (Fig 3C and Appendix Fig S7). Notably, the strongest CENATAC interactor (TXNL4B) was also the gene that showed the

most significant co-occurrence with CENATAC in eukaryotic species (Fig 3A). To determine whether CENATAC preferentially associates with major or minor spliceosome components, we analyzed CENATAC co-immunoprecipitations by Northern blot analysis. This revealed a significant enrichment for the minor spliceosome-specific U4atac and U6atac snRNAs (Fig 3D and E and Appendix Fig S8). CENATAC's association with the minor spliceosome was further supported by glycerol gradient analyses of HeLa nuclear extract preparations, which showed co-migration of CENATAC with U6atac snRNP, U4atac/U6atac di-snRNP, and U4atac/U6atac.U5 tri-snRNP complexes (Fig 3F). Together, these data validate CENATAC as a bona fide functional component of the minor spliceosome and as the first identified protein component that is specific to the U4atac/U6atac and U4atac/U6atac.U5 snRNP complexes (Fig 3G).

The role of *CENATAC* in minor spliceosome function was further supported by the presence of evolutionarily conserved competing major (U2-type) and minor (U12-type) 5′ splice sequences (5′ss) in animals, that are predicted to generate productive and unproductive *CENATAC* mRNAs, respectively (Appendix Fig S9A). This configuration is indicative of an autoregulatory circuit that is conceptually

similar to the previously reported autoregulation of the minor spliceosome proteins 48K and 65K (Verbeeren *et al*, 2010; Turunen *et al*, 2013b). In agreement with this, impaired minor spliceosome function, such as in Taybi–Linder syndrome (TALS/MOPD1, OMIM: 210710) patients, leads to a significant increase in the use of major

5′ss and upregulation of *CENATAC* mRNA levels (Cologne *et al*, 2019). Notably, evidence for a similar autoregulatory circuit is also present in plants (Appendix Fig S9B), where retention or splicing of a minor intron results in productive or unproductive *CENATAC* mRNA, respectively.

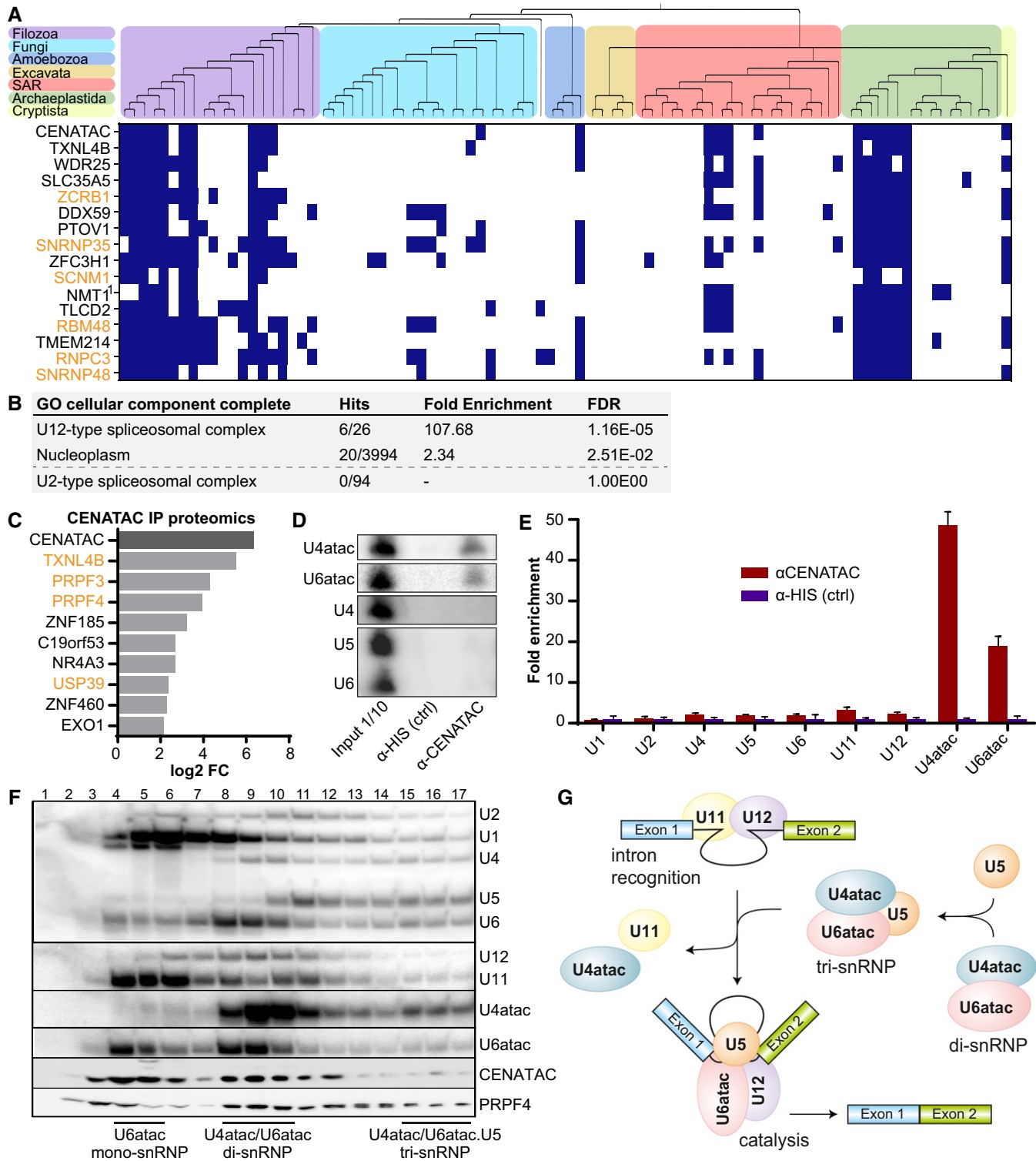

**Figure 3.**

◀

**Figure 3. CENATAC is a novel component of the minor spliceosome.**

A Phylogenetic profiles (presences (blue) and absences (white)) of the top 15 genes co-occurring with *CENATAC* in 90 eukaryotic species. Top: phylogenetic tree of the eukaryotic species (see Table EV1) with colored areas for the eukaryotic supergroups. [1]For *NMT1*, no human ortholog was found and instead the *Arabidopsis thaliana* ortholog is depicted. Genes associated with the minor (U12-dependent) spliceosome are depicted in orange. See also Table EV2.

B GO term analysis of the genes co-occurring with *CENATAC* as in (A) with a correlation score of > 0.5. Note: The amount of hits and fold enrichment score were manually changed to accommodate the recently discovered SCNM1 (Bai *et al*, 2021).

C Graph of fold changes in proteins enriched (*P*-value < 0.05) in proteomics analysis of CENATAC vs. control co-immunoprecipitations of HeLa$^{EGFP-CENATAC}$ cells (three biological replicates). Splicing factors are depicted in orange. See also Appendix Fig S7.

D, E Examples (D) and quantification (E) of Northern blot analyses of minor (U6atac, U4atac, U11, and U12) and major (U2, U1, U4, U5, and U6) spliceosome snRNAs in HeLa$^{EGFP-AID-CENATAC}$ cells (three biological replicates, normalized to the control). See also Appendix Fig S8.

F Glycerol gradient (10–30%) analysis of HeLa S3 nuclear extracts. snRNAs were detected by Northern blot analysis, proteins (CENATAC and PRPF4) by Western blot. Locations of the U6atac mono-snRNP, U4atac/U6atac di-snRNP, and U4atac/U6atac.U5 tri-snRNP are indicated.

G Schematics showing key assembly stages in minor intron splicing and minor tri-snRNP assembly: intron recognition (A complex) and the catalytic spliceosome (C complex). For simplicity, several stages of spliceosome assembly are omitted, such as the pre-B complex, which consists of the intron recognition complex together with the tri-snRNP before architectural changes lead to the exclusion of U11 to give rise to the B complex, after which subsequent architectural changes lead to the exclusion of U4atac to give rise to the B$^{ACT}$ complex, which is a precursor stage for the catalytically active C complex depicted in this figure (Turunen *et al*, 2013b).

Data information: In (C), data are presented as fold change of the mean log 2-transformed LFQ intensity. In (E), data are presented as mean ± SD.

## Minor intron splicing defects in *CENATAC* mutant cells correlate with mitotic defects

In agreement with our finding that CENATAC is a novel minor spliceosome component, splicing of several minor introns was impaired upon depletion of CENATAC in HeLa$^{EGFP-AID-CENATAC}$ cells, whereas up- or downstream major introns were unaffected (Fig 4A and B). This was true also for MVA patient cells and DLD-1 cells expressing MVA mutant CENATAC (Fig 4B and Appendix Fig S10). Importantly, the splicing defect of minor introns in CENATAC-depleted cells was fully rescued by re-expression of wild-type but not MVA mutant alleles (Fig 4A). Similar to the mitotic phenotype, the expression of the disease alleles and mutants lacking motifs 3 and 4 exacerbated the splicing defect, whereas mutations in the zinc fingers and removal of motifs 1 and 2 partially rescued it (Fig 4A and Appendix Fig S10). Notably, the extent of the splicing defect strongly correlated with the extent of the mitotic phenotype for all mutations (Fig 4C), supporting the possibility that impaired minor spliceosome function and the chromosome congression phenotype are causally linked. To further investigate this, we depleted ZRSR2, a component of the U11/U12 di-snRNP functioning in 3′ss recognition of minor introns (see Fig 3G), in both HeLa$^{EGFP-AID-CENATAC}$ and DLD-1 cells. Similar to the depletion of CENATAC, this caused significant minor IR (Fig 4D) and a chromosome congression defect (Fig 4E). We therefore consider it likely that the chromosome congression phenotype is a secondary effect of impaired minor spliceosome function.

## CENATAC promotes splicing of A-type minor introns

Our discovery of reduced minor spliceosome function in a constitutional aneuploidy syndrome raised the question of which introns and transcripts were affected by CENATAC malfunction. To investigate this, we compared the transcriptomes of CENATAC-depleted HeLa$^{EGFP-AID-CENATAC}$ cells to those of control-depleted and parental cell lines. The resulting RNAseq dataset was analyzed for changes in IR using IntEREst (Oghabian *et al*, 2018) and for alternative splicing (AS) using Whippet (Sterne-Weiler *et al*, 2018). In agreement with our RT–PCR-based observations (Fig 4), this analysis confirmed the significant retention of minor but not major introns after

CENATAC depletion (Fig EV3). Surprisingly, it also uncovered a remarkable enrichment for a specific subclass of minor introns: While only 24% of G-type introns (with GT-AG, GT-AT, GT-TG, GC-AG terminal dinucleotides) were affected by CENATAC depletion, virtually all (92%) of the A-type introns (with AT-AC[1], AT-AA, AT-AG, or AT-AT terminal dinucleotides) showed increased retention or activation of alternative major splice sites (cryptic or annotated), or both (Fig 5A and B, Appendix Fig S11, and Dataset EV1). For comparison, we carried out the same analysis on a previously published dataset derived from myelodysplastic syndrome (MDS) patients carrying somatic mutations in the gene encoding for the U11/12-di-snRNP subunit *ZRSR2* (Madan *et al*, 2015). This dataset showed a nearly identical response for A- and G-type introns (Fig 5B and Dataset EV2). Whereas depletion of CENATAC or mutations in *ZRSR2* led to an average increase of approximately 36 and 19% in retention of A-type introns, respectively (average $\Delta\Psi_{CENATAC} = \sim 0.36$ and average $\Delta\Psi_{ZRSR2} = \sim 0.19$), G-type introns were only strongly affected by *ZRSR2* mutations (average $\Delta\Psi_{CENATAC} = \sim 0.07$ and average $\Delta\Psi_{ZRSR2} = \sim 0.19$, Figs 5B–D and EV3). Importantly, the effect on A-type introns was specific to minor introns as none of the 85 major AT-AC introns or related subtypes responded to CENATAC depletion (Fig 5E and Dataset EV1). The same subtype-specific effect on minor intron splicing was also observed in *CENATAC* mutant MVA patient lymphoblasts (Fig 5F, Appendix Fig S12A, and Dataset EV3), in which the affected introns correlated strongly with those affected by CENATAC depletion (Fig 5G, Appendix Fig S12B, and Dataset EV4). Notably, the strongly affected transcripts did not include any of the genes associated with MVA but did contain various mitotic regulators (Appendix Fig S13, Datasets EV1, EV5, and EV6).

## A-type minor introns are spliced less efficiently

We next wished to understand the selectivity of CENATAC-dependent splicing for A-type minor introns. The observation that also some G-type introns were affected by CENATAC depletion (Fig 5D) argued against a direct interaction between CENATAC and intron terminal nucleotides. Moreover, A- and G-type introns that were strongly affected by CENATAC depletion had higher basal levels of IR in control conditions compared with those unaffected by

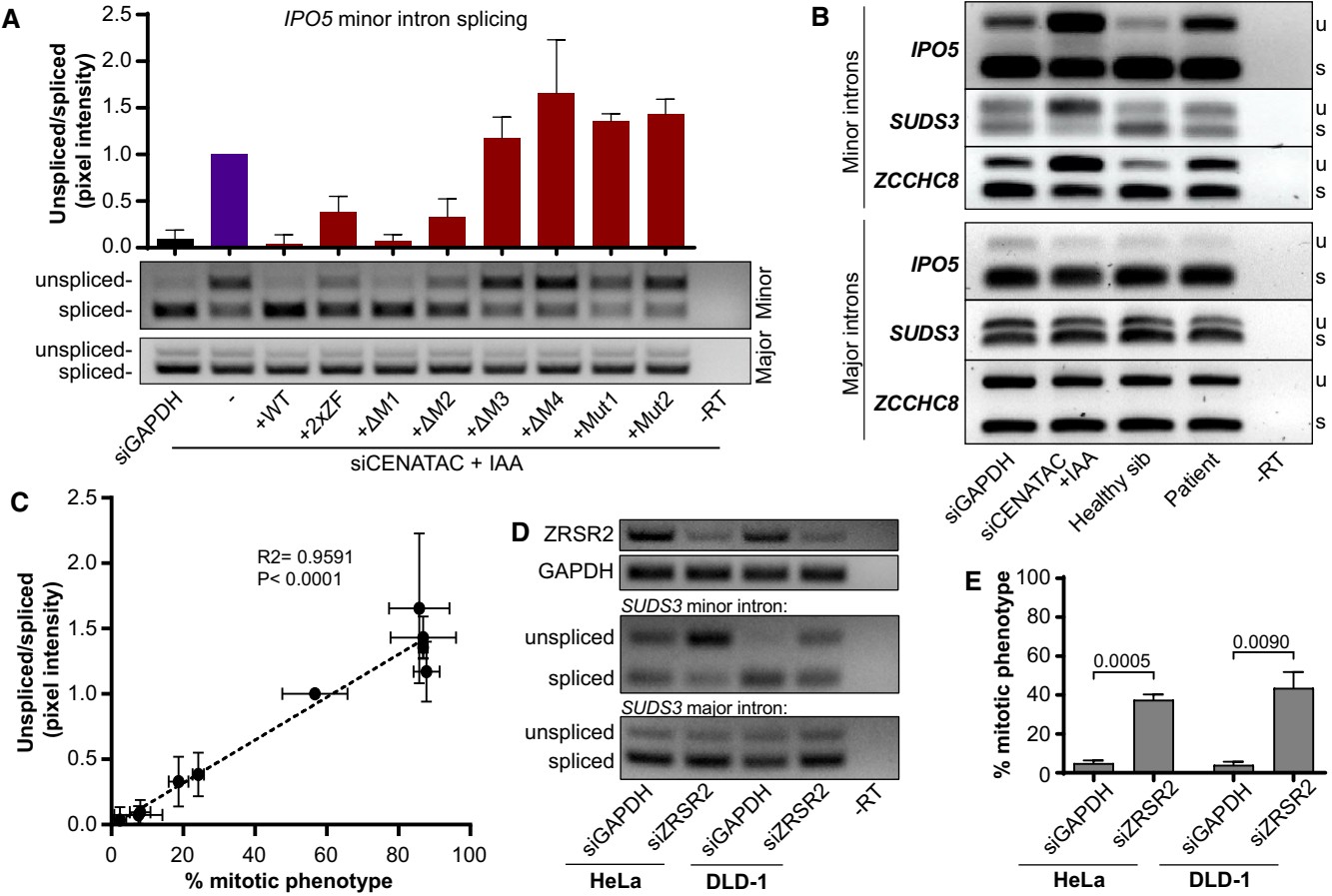

**Figure 4. Minor intron splicing defects in *CENATAC* mutant cells fully correlate with mitotic defects.**

A  RT–PCR (middle panel) and quantification (upper panel) of *IPO5* minor intron 21 (U12-type) splicing on RNA extracted from HeLa[EGFP-AID-CENATAC] cells treated as in Fig 2C (three biological replicates, normalized to siCENATAC + IAA). The bottom panel shows RT–PCR analysis of IPO5 major intron 19 (U2-type).

B  RT–PCRs of *IPO5, SUDS3,* and *ZCCHC8* minor and adjacent major introns on RNA extracted from patient lymphoblasts or HeLa[EGFP-AID-CENATAC] cells depleted of GAPDH or CENATAC as indicated. u, unspliced; s, spliced.

C  For each condition in (A), minor intron splicing (Fig 4A) was plotted against the percentage of cells showing a mitotic phenotype (Fig 2C). $R^2$ and *P*-values are provided for the linear regression trendline (dotted line).

D  RT–PCRs of *ZRSR2* and *GAPDH* (to visualize ZRSR2 knockdown efficiency, upper) and RT–PCRs of *SUDS3* minor intron 7 (middle) and major intron 5 (lower) on RNA extracted from HeLa[EGFP-AID-CENATAC] or DLD-1 cells depleted of GAPDH or ZRSR2 as indicated.

E  Quantification of mitotic defects as in Appendix Fig S6 of H2B-mNeon-expressing HeLa[EGFP-AID-CENATAC] or DLD-1 cells treated as in (D) (three biological replicates, > 103 cells in total per condition).

Data information: In (A, E), data are presented as mean ± SEM. *P*-values were calculated with unpaired Student's *t*-tests. In (C), the *P* and $R^2$ values were calculated with a linear regression analysis.

CENATAC depletion (Fig 6A). This suggested that CENATAC predominantly promotes splicing of minor introns that are normally spliced less efficiently and that A-type introns as a group belong to this category. To test this hypothesis, we engineered the widely used P120 minigene (Hall & Padgett, 1996) to contain two tandem competing A- or G-type splice sites in all possible configurations (Fig 6B). Significantly, minor GT-AG splice sites were strongly preferred over AT-AC sites when in direct competition (Fig 6B and C), and they also outcompeted a unique GC-AG splice site that was significantly affected by CENATAC depletion (Figs 5D and EV4, *LZTR1*). We thus conclude that CENATAC promotes splicing of minor introns that are recognized or spliced less efficiently, most prominently A-type minor introns.

## Discussion

In this work, we have uncovered a novel link between the minor spliceosome and defects in chromosome segregation in human cells. Using patient exome sequencing, evolutionary co-occurrence analysis, and biochemistry, we identified CENATAC as a novel protein component of the U4atac/U6atac di-snRNP and U4atac/U6atac.U5 tri-snRNP complexes that are necessary for the formation of the catalytically active minor spliceosome. Our RNAseq analyses of *CENATAC*-mutant MVA patient cells and CENATAC-depleted HeLa[EGFP-AID-CENATAC] cells revealed widespread defects in minor intron splicing, particularly IR, but also cryptic splice site activation, indicating that CENATAC is required for proper functioning of the

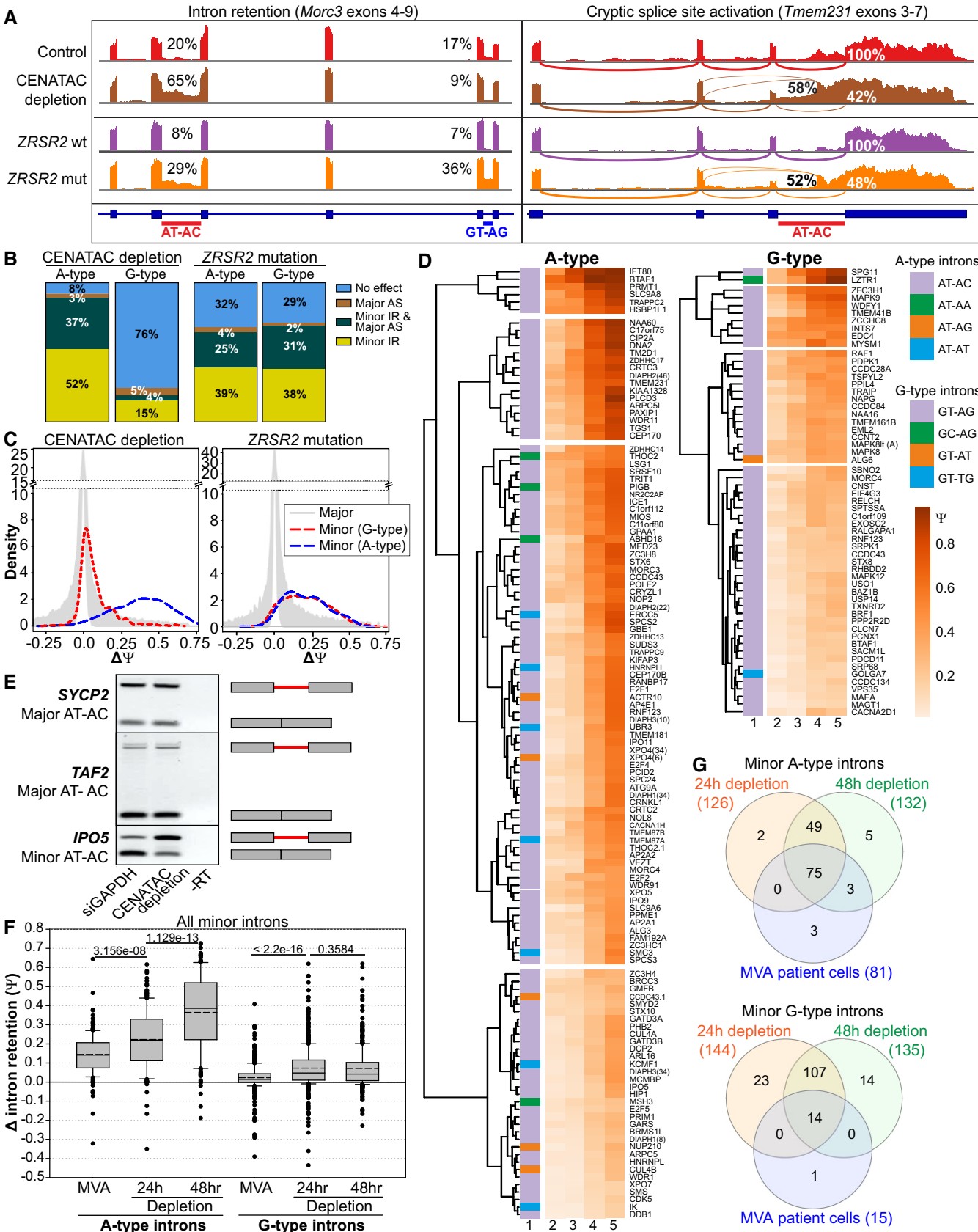

**Figure 5.**

◄

**Figure 5.   CENATAC promotes splicing of A-type minor introns.**

A   Sashimi plots showing the effect of CENATAC depletion (48hr, HeLa[EGFP-AID-CENATAC] cells) or *ZRSR2* mutations (MDS patient cells) on AT-AC and GT-AG intron retention (left panel, *Morc3)*, and cryptic splice site activation (right panel, *Tmem231*). CENATAC control represents the parental unedited HeLa cell line.

B   Transcriptome-wide statistics of CENATAC depletion (48hrs) and *ZRSR2* mutations on G- and A-type minor intron retention (U12 IR) and cryptic major splice site activation (U2 AS). Only introns showing at least 5 exon–exon junctions reads were included. For U12 IR, a statistical cutoff of $P_{adj} < 0.05$ was used. For U2 AS, the probability cutoff of $P_r > 0.9$ was used.

C   Density plots showing differences in intron retention ($\Delta\Psi$) distribution after CENATAC depletion (48 h) or in samples with *ZRSR2* mutations.

D   Hierarchical clustering of A- and G-type intron retention in the unedited parental cell line treated with siGADPH for 48h (column 2), or in the HeLa[EGFP-AID-CENATAC] cell line treated with siGADPH for 48h (column 3) or with auxin and siCENATAC for 24 or 48 h (columns 4 and 5, respectively). Only introns showing a $P_{adj} < 0.05$ and $\Delta\Psi > 0.1$ in either the 24-h or 48-h depletion sample were included in the analysis. The A-type and G-type intron terminal dinucleotide subtypes are indicated with different colors in the first column. In case the gene contained multiple introns of the same type, the intron number is indicated in parentheses.

E   RT–PCR of major (U2-type) AT-AC introns in *SYCP2* (intron 5) and *TAF2* (intron 1), and minor (U12-type) AT-AC intron in *IPO5* (intron 21) on RNA extracted from HeLa[EGFP-AID-CENATAC] cells depleted of GAPDH or CENATAC. Schematic representations of unspliced/spliced PCR products are depicted on the right.

F   Δ_intron_retention values for the MVA patient cell dataset (compared with the healthy sibling) using all (significant and not significant) minor A-type introns ($n = 179$ for the depletion and $n = 177$ for the MVA patient datasets) and minor G-type introns ($n = 441$ for the depletion and $n = 446$ for the MVA patient datasets). Only introns with on average at least 5 intron mapping reads were used in the analysis. See also Appendix Fig S12.

G   Venn diagram analysis of the MVA patient cell and HeLa[EGFP-AID-CENATAC] CENATAC depletion datasets. A- and G-type minor introns showing statistically significant intron retention in each dataset were used. See also Dataset EV4.

Data information: In (F), data are presented as median (solid line) and mean (dashed line) inside the boxes. The boundaries of the boxes indicate 25th and 75th percentiles. Whiskers indicate the 90th and 10th percentiles. *P*-values were calculated with two-sided Mann–Whitney rank-sum tests. The CENATAC depletion dataset consists of three biological replicates, the *ZRSR2* mutation (MDS) dataset of eight biological replicates, and the MVA dataset of four biological replicates.

minor spliceosome. Unexpectedly, IR in CENATAC-depleted cells was strongly biased for A-type minor introns, which is a subtype that is defined by AT-AN dinucleotide splice sites. This intron subtype-specific function is unique among the minor spliceosome components and correlated tightly with mitotic fidelity. Furthermore, depletion of the minor spliceosome component ZRSR2 likewise caused a chromosome congression defect. Minor intron subtype mis-splicing is therefore likely responsible, possibly in conjunction with centriolar defects (Wang *et al*, 2019), for the inefficient chromosome congression in *CENATAC*-mutant cells and for the aneuploidies observed in the two MVA siblings described in this study.

The minor spliceosome was originally thought to splice only introns with AT-AC termini (Hall & Padgett, 1994, 1996; Tarn & Steitz, 1996). Only later, it was shown that there are also major AT-AC introns and that most of the minor introns in fact have GT-AG termini (Dietrich *et al*, 1997; Sharp & Burge, 1997; Wu & Krainer, 1997). Additionally, minor introns have infrequent variations in the 3′ terminal nucleotide (Levine & Durbin, 2001; Dietrich *et al*, 2005), thus giving rise to the AT-AN and GT-AN classes of minor introns, here referred to as A- and G-type introns, respectively. Significantly, in all known minor spliceosome diseases for which comprehensive transcriptome data are available, splicing defects are roughly uniformly distributed between the A- and G-type introns (Argente *et al*, 2014; Madan *et al*, 2015; Merico *et al*, 2015; Cologne *et al*, 2019) (Fig 5A–C, *ZRSR2* mutation). The selective A-type IR phenotype of CENATAC can therefore not solely be explained by a general loss of minor spliceosome function but instead suggests that CENATAC has a unique function in promoting the splicing of A-type minor introns. Nonetheless, our observation that also a subset of G-type introns was retained upon CENATAC depletion renders it unlikely that CENATAC directly recognizes the 5′ adenosine of A-type introns. Instead, our competition data, and the observed elevated IR baseline of the affected introns, suggest that a subset of minor introns with reduced intrinsic splicing activity (consisting not only of A-type introns but also of a subset of G-type introns) may be particularly dependent upon CENATAC activity.

What could be the molecular function of CENATAC? Given its participation in di- and tri-snRNP complexes and potential

selectivity for 5′ss identity, CENATAC may function during or after the transition from initial intron recognition (A complex) to pre-catalytic spliceosome (B complex; Fig 3G) and may for instance participate in 5′ss recognition analogous to U11-48K protein in the A complex (Turunen *et al*, 2008). Of particular interest are the two well-conserved C-terminal motifs (M3 and M4), whose deletion caused severe impairment of minor intron splicing (Figs 1 and 4). Although we have not been able to uncover their function based on sequence similarity to other motifs, our data argue they are crucial for CENATAC's role in minor intron splicing. Detailed mechanistic understanding will require cryo-EM structures of relevant minor spliceosome assembly stages. Unlike the major spliceosome, of which high-resolution structures are available throughout the entire spliceosome assembly/disassembly cycle (Wilkinson *et al*, 2020), of the minor spliceosome only a single high-resolution cryo-EM structure is available of the catalytically activated form (B[ACT] complex) (Bai *et al*, 2021). This structure does not contain CENATAC, nor its main interactor TXNL4B (Fig 3). Even though the major spliceosome does not carry an obvious functional analog of the CENATAC protein, the major spliceosome TXNL4A/Dim1 protein is a paralog of TXNL4B, both at sequence and structural levels (Jin *et al*, 2013). High-resolution structures of both yeast and human spliceosomal B complexes have placed TXNL4A/Dim1 in close proximity of the 5′ss and suggested a role in 5′ss recognition (Wan *et al*, 2016; Bertram *et al*, 2017). Assuming that the minor spliceosome B complex shares the molecular architecture with its major spliceosome counterpart, this could place CENATAC with TXNL4B near the 5′ss to participate in the recognition event. Furthermore, as both proteomics and structural work have shown that TXNL4A/Dim1 is released from the major spliceosome during the transition from B to B[ACT] complex (Schmidt *et al*, 2014; Bertram *et al*, 2017; Haselbach *et al*, 2018), it is possible that both TXNL4B and CENATAC may similarly detach from the minor spliceosome prior to B[ACT] complex formation.

Given that A-type minor intron host genes and the locations of these introns within the host gene are evolutionarily highly conserved among metazoan species, the selectivity of CENATAC in splicing raises the possibility that minor intron subtypes are part of a conserved but unexplored regulatory mechanism for gene

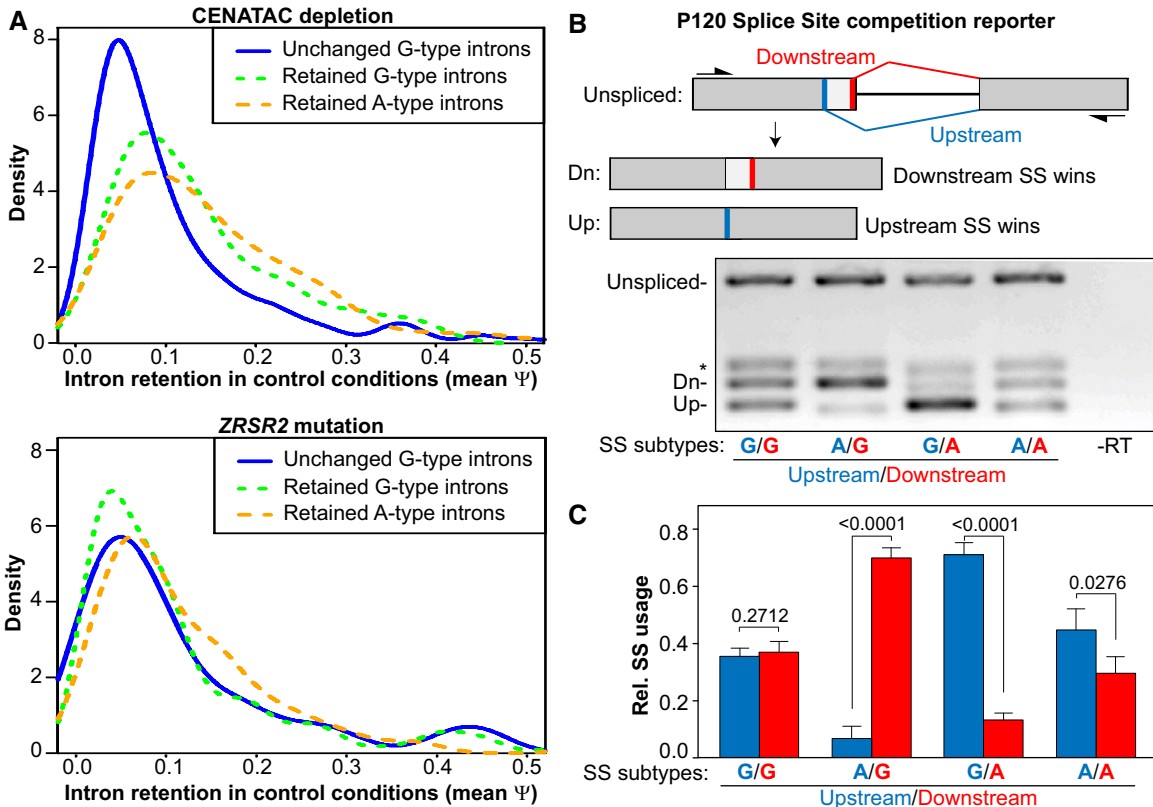

**Figure 6. A-type minor introns are spliced less efficiently.**

A   Density plots showing intron retention (Ψ) values in the HeLa unedited parental control cell line (depleted of GAPDH) of A- and G-type minor introns that were either unchanged or retained after CENATAC depletion (top) or *ZRSR2* mutation (bottom). The median psi values of the retained G- and A-type introns are significantly higher (Psi = 0.130 and Psi = 0.115, respectively; $P < 0.01$, Mann–Whitney rank-sum test) compared with the unchanged introns (Psi = 0.081) in the CENATAC depletion dataset.

B   RT–PCR P120 reporter assay (Hall & Padgett, 1996) to measure the relative usage of A-type (AT-AC) and G-type (GT-AG) 5′ splice sites in direct competition. Upper: schematic diagram showing the overall architecture of the reporter construct with its down- and upstream splice site (thick red and blue bars, respectively) and the products created by splicing (Dn and Up, respectively). Lower: RT–PCRs of the reporter with A- or G-type splice sites in the down- or upstream positions as indicated below the gel. SS, splice site. *PCR product after use of a cryptic major splice site (not shown in the schematic).

C   Quantification of relative splice site usage of A- and G-type splice sites in (B) (three biological replicates).

Data information: In (C), data are presented as mean ± SEM. *P*-values were calculated with unpaired Student's *t*-tests. The CENATAC depletion and *ZRSR2* mutation (MDS) datasets (panel (A)) consist of three and eight biological replicates, respectively.

expression. CENATAC undergoes reversible modifications (acetylation and phosphorylation) (Wang *et al,* 2019), which may provide the means to regulate its activity (also) in the minor spliceosome.

Presently, all mutations associated with MVA have been mapped to genes that are known regulators of chromosome segregation. Our discovery of disease-causing mutations in *CENATAC* extends this list for the first time with a mRNA splicing factor. Although a recent study showed that CENATAC regulates centriole duplication (Wang *et al,* 2019), we were unable to verify this. Instead, our data argue that chromosomal instability by CENATAC malfunction may instead be the result of a primary defect in splicing of A-type minor introns. Nevertheless, it remains possible that CENATAC can also promote high-fidelity chromosome segregation more directly, as has been suggested for several other proteins involved in splicing (Montembault *et al,* 2007; Pellacani *et al,* 2018; Somma *et al,* 2020).

Strikingly, the clinical phenotype of *CENATAC* mutant MVA strongly resembles that of MOPD1/TALS, Roifman and Lowry–Wood syndromes, which are caused by mutations in the U4atac

snRNA component of the minor spliceosome. Patients with these syndromes likewise present with microcephaly, developmental delay, and retinal abnormalities (Farach *et al,* 2018). No aneuploidies were reported (Hallermayr *et al,* 2018; Wang *et al,* 2018), but karyotype analyses were not performed for the majority of patients. It will therefore be of interest to examine whether aneuploidies occur in some of these patients, and whether (and to what extent) the affected transcripts and the splicing defect differ between MVA and these syndromes.

Although the depletion of both ZRSR2 and CENATAC caused a chromosome congression defect in mitosis (Fig 4), patients with *ZRSR2* mutations are clinically different from MVA patients with *CENATAC* mutations. This difference is most likely related to differences in their splice targets, such as the G-type minor introns that are differentially affected by depletion of CENATAC vs ZRSR2 (Fig 5). Mutations in *ZRSR2* are associated with MDS and clonal cytopenias of unknown significance (CCUS) (Madan *et al,* 2015; Fleischman *et al,* 2017). In line with ZRSR2's mitotic phenotype,

various stable aneuploidies were observed in MDS and CCUS patients with *ZRSR2* mutations (Madan *et al*, 2015; Fleischman *et al*, 2017; Hosono, 2019), though it is unclear whether mitotic defects contribute to these disease phenotypes. It would be of interest to investigate whether mitotic defects negatively impact erythropoiesis in these patients.

# Materials and Methods

## Samples

The MVA exome analyses were approved by the London Multicentre Research Ethics Committee (05/MRE02/17). Appropriate consent was obtained from patients and/or parents as applicable and the experiments conformed to the principles set out in the WMA Declaration of Helsinki and the Department of Health and Human Services Belmont Report. DNA was extracted from whole blood using standard protocols. RNA was extracted from EBV-transformed lymphoblastoid cell lines (LCLs) using the RNeasy Mini Kit protocol (Qiagen).

For the functional experiments, the following patient LCLs were used: ID_5728_1 (patient, biallelic *CENATAC* (*CCDC84*) mutations, ECACC ID: FACT5728DLB), ID_5728_3 (sibling, no *CENATAC* mutations, ECACC ID: FACT5728KC), ID_5728_4 (father, monoallelic *CENATAC* mutation, ECACC ID: FACT5728GLB), and ID_5728_5 (mother, monoallelic *CENATAC* mutation, ECACC ID: FACT5728ALB).

Lymphoblastoid cell lines were cultured in RPMI supplemented with 15% fetal bovine serum (FBS), 100 μg/ml penicillin/streptomycin, and 2 mM alanyl glutamine. Cells expressing H2B-mNeon were created by lentiviral transduction, using standard procedures. Imaging of LCLs was performed as previously described (Yost *et al*, 2017).

Exome sequencing, alignment and variant calling, reference data sets, PTV prioritization method, recessive analysis, and Sanger sequencing: as previously described (Yost *et al*, 2017).

## cDNA analysis of *CENATAC* (*CCDC84*) mutations

We synthesized cDNA using the ThermoScript RT–PCR System (Life Technologies) with random hexamers and 1 μg of total RNA. We amplified the mutation regions using cDNA-specific primers and sequenced the PCR products as described above. Primer sequences are available on request.

## Conservation logos

Hidden Markov model (HMM) profiles were created from iterative jackhmmer searches (Potter *et al*, 2018) (version: HMMER3/f [3.1b2 | January 2014]) with CENATAC's protein sequence against the sequences of all metazoan species within the UniProt database. In-between successive iterations, non-CENATAC sequences were manually removed. Logos were created using Skylign (Wheeler *et al*, 2014); letter height: information content above background.

## Immunoblots

For Western blot samples, cells were treated as indicated and lysed in Laemmli lysis buffer (4% SDS, 120 mM Tris pH 6.8, and 20%

glycerol). Lysates were processed for SDS–polyacrylamide gel electrophoresis and transferred to nitrocellulose membranes. Immunoblotting was performed using standard protocols. Visualization of signals was performed on an Amersham Imager 600 scanner using enhanced chemiluminescence. Primary antibodies used were rabbit anti-CENATAC (CCDC84; Sigma, HPA071715) and mouse anti-Tubulin (Sigma; T5168). Secondary antibodies used were goat anti-mouse HRP (170-6516) and goat anti-rabbit HRP (170-6515), both obtained from Bio-Rad.

## Cell culture

HeLa T-REx Flp-In osTIR-9Myc::NEO cells (gift from Andrew Holland) were cultured in DMEM high glucose supplemented with 10% Tet-approved FBS, 100 μg/ml penicillin/streptomycin, and 2 mM alanyl glutamine. DLD-1 cells (ATCC CCL-221) were cultured in DMEM/F-12 supplemented with 10% Tet-approved FBS, 100 μg/ml penicillin/streptomycin, and 2 mM alanyl glutamine. HeLa S3 cells (a kind gift from Dr. Joan Steitz) were cultured in suspension in 1640 RPMI supplemented with 10% FBS, 2 mM glutamine, and 100 μg/ml penicillin/streptomycin. Stable expression of H2B-mNeon was done by lentiviral transduction using standard procedures. All cell lines were regularly tested and at all times found to be mycoplasma-free.

## Creation of HeLa$^{EGFP-AID-CENATAC}$ and HeLa$^{EGFP-CENATAC}$ cell lines

HeLa$^{EGFP-AID-CENATAC}$ and HeLa$^{EGFP-CENATAC}$ cell lines were derived from HeLa T-REx Flp-In osTIR-9Myc::NEO and HeLa T-REx Flp-In, respectively. Tagging of the endogenous locus of CENATAC was done according to the scCRISPR protocol (Arbab *et al*, 2015) using the Protospacer, HDR_insert, and HDR_ext primers in Table EV3. pcDNA5-FRT-TO-EGFP-AID (Addgene, 80075) was used as template for both the EGFP-AID and EGFP tags. Cells were transfected with Lipofectamine LTX using standard procedures and subsequently FACS-sorted (single cells) based on EGFP expression. Endogenous tagging was confirmed by PCR (using the Genomic primers, Appendix Fig S3A) and immunoblotting of CENATAC protein (Appendix Fig S3B and C).

## Viral plasmids, cloning, and viral production

For lentiviral re-expression of CENATAC variants, first pcDNA5 PURO FRT TO EGFP-AID-CENATAC was created by cloning CENATAC cDNA derived from HeLa cells into empty pcDNA5-FRT-TO-EGFP-AID (Addgene, 80075) using the cDNA PCR primers in Table EV3 and digestion of both the PCR product and the plasmid with NotI/ApaI. The CENATAC cDNA was subsequently cloned into pcDNA5 PURO FRT TO containing a LAP-tag to create pcDNA5 PURO FRT TO LAP-CENATAC by Gibson assembly (Gibson *et al*, 2009) with the PCR primers Gibson1 and Gibson2. Mutagenesis was then performed to make this construct resistant to CENATAC siRNA treatment (CCDC84; Dharmacon, J-027240-07) by Gibson assembly with PCR primers Gibson3. Next, in the siRNA-resistant construct, CENATAC wild-type cDNA was mutated to Mut1 (primers Gibson4), Mut2 (Gibson5), 2xZF (Gibson6; two consecutive rounds of cloning), Δ1 (Gibson7), Δ2 (Gibson8), Δ3 (Gibson9), or Δ4 (Gibson10) by Gibson assembly. Lentiviral CENATAC iresRFP constructs

were derived from a lentiviral construct encoding fluorescently tagged histone 2B (H2B) and a puromycin-resistant cassette (pLV-H2B-mNeon-ires-Puro) (Drost *et al*, 2015). First, the fluorescently tagged H2B was substituted by CENATAC derived from pcDNA5 PURO FRT TO LAP-CENATAC (see above) by Gibson assembly with PCR primers Gibson11 and digestion by AscI/NheI. Next, the puromycin-resistant cassette was substituted by tagRFP by Gibson assembly with PCR primers Gibson12. Finally, all siRNA-resistant variants of CENATAC were cloned from their respective pcDNA5 PURO FRT TO LAP-CENATAC plasmids into pLV CENATAC ires-tagRFP by Gibson assembly with PCR primers Gibson13 and PstI digestion of the plasmid. Virions were generated by transient transfection of HEK 293T cells with the transfer vector and separate plasmids that express Gag-Pol, Rev, Tat, and VSV-G. Supernatants were clarified by filtration.

## Immunoprecipitation

For each sample, a full 10-cm plate of HeLa$^{EGFP-AID-CENATAC}$ cells was used, treated as indicated (Appendix Fig S3C). The cells were lysed in ice-cold lysis buffer (50 mM Tris pH 7.5, 150 mM NaCl, 2% NP-40, 0.1% deoxycholate, proteasome inhibitors) and treated with benzonase for 15 min at 4°C. After centrifugation, the supernatant was incubated with beads (GFP-Trap, Chromotek) for 2.5 h at 4°C and washed three times with ice-cold lysis buffer. The samples were finally eluted in Laemmli sample buffer.

## Live cell imaging analysis of mitotic fidelity

Lymphoblastoid cell lines were imaged as previously described (Yost *et al*, 2017). siRNA transfections (RNAiMAX, Thermo Fisher) in HeLa$^{EGFP-AID-CENATAC}$ (40 nM siRNA) and DLD-1 cells (50 nM siRNA) were done against CENATAC (CCDC84; Dharmacon, J-027240-07), GAPDH (Dharmacon, D-001830-01-05), or ZRSR2 (Sigma, SASI_Hs02_00338940). In the case of CENATAC depletion in HeLa$^{EGFP-AID-CENATAC}$ cells, transfections were done in the presence of 1 mM 3-indoleacetic acid (IAA) or ethanol (IAA vehicle) for 24 h in a 24-well plate before the cells were re-plated to eight-well ibidi μ-slides with 2 mM thymidine (for early S-phase synchronization) and 100 μl lentivirus for CENATAC re-expression. After 18 h, the cells were released from thymidine for 6 h and imaged in CO$_2$-independent medium in a heated chamber (37°C), while air-tight-sealed in the well plate with parafilm. These cells were therefore imaged ~ 48 h after siRNA-mediated knockdown of CENATAC and ~ 24 h after lentivirus addition. For CENATAC depletion and re-expression in DLD-1 cells, the lentivirus (150 μl) was immediately added together with the siRNA treatment (instead of 24 h later together with the 2 mM thymidine). These cells were therefore imaged ~ 48 h after siRNA-mediated knockdown of CENATAC and ~ 48 h after lentivirus addition. For the experiments in Figs 2D and EV2A, the cells were additionally incubated with 200 nM SiR-tubulin dye (Spirochrome) for 6 h prior to imaging to facilitate visualization of the mitotic spindle. For the depletion of ZRSR2 in both HeLa$^{EGFP-AID-CENATAC}$ and DLD-1 cells, the cells were re-plated to 8-well ibidi μ-slides with 2 mM thymidine 48 h (instead of 24 h) after transfection and therefore imaged ~ 72 h after siRNA-mediated knockdown of ZRSR2. Images were acquired every 3 or 5 min at 1 × 1 binning in 7 × 2.5 μm z-stacks (RFP as in Appendix Figs S4

and S6 was imaged in only 1 z-stack per position) and projected to a single layer by maximum intensity projection using NIS-Elements Software 4.45. Imaging was performed with a Nikon Ti-Eclipse wide-field microscope equipped with an Andor Zyla 4.2 sCMOS Camera, 40× oil objective NA 1.3 WD 0.2 mm, and Lumencor SPECTRA X light engine. Analysis of these experiments was carried out with ImageJ software. When applicable, cells re-expressing CENATAC variants were identified through co-expression of cytosolic RFP (via ires-tagRFP); RFP-negative cells were omitted from the quantifications (Appendix Figs S4 and S6).

## Immunofluorescence imaging

After treating the cells with siRNAs and IAA (see above) for 24 h in a 24-well plate, the cells were re-plated on round 12-mm coverslips and treated with 2 mM thymidine (for early S-phase synchronization) for 24 h. 10 h after release, MG132 was added for 45 min after which the cells were pre-extracted with 0.1% Triton X-100 in PEM (100 mM PIPES pH 6.8, 1 mM MgCl$_2$, and 5 mM EGTA) for ± 60 s. After 60 s, 4% paraformaldehyde was added on top of the PEM in a 1:1 ratio (400 μl each) for 20 min to fixate the cells. The coverslips were subsequently washed twice with PBS and blocked with 3% BSA in PBS for 16 h at 4°C, incubated with primary antibodies for 2 h at room temperature, washed three times with PBS containing 0.1% Triton X-100, and incubated with secondary antibodies for 1 h at room temperature. Coverslips were then washed four times with PBS/0.1% Triton X-100 and mounted using ProLong Gold Antifade with DAPI (Molecular Probes). All images were acquired on a deconvolution system (DeltaVision Elite; Applied Precision/GE Healthcare) equipped with a 100×/1.40 NA UPlanSAPO objective (Olympus) using Softworx 6.0 software (Applied Precision/GE Healthcare). The images are maximum intensity projections of deconvoluted stacks. Random pro-metaphase and metaphase cells were selected, and centrioles were counted by hand. Primary antibodies used were rabbit anti-Centrin1 (Abcam, ab101332, 1/500) and mouse anti-Tubulin (Sigma, T5168, 1/10,000). Secondary antibodies used were goat anti-mouse 647 (A21236) and goat anti-rabbit 568 (A11036), both obtained from Thermo Fisher.

## Co-evolution analysis

First, a phylogenetically diverse set of complete eukaryotic-predicted proteomes was utilized. This set was previously compiled to contain the protein sequences of 90 eukaryotic species (Hooff *et al*, 2017; preprint: van Wijk & Snel, 2020). These species were selected based on their representation of eukaryotic diversity. If available, we selected two species per clade and model organisms were preferred over other species. If multiple proteomes or proteomes of different strains were available, the most complete proteome was selected. When multiple splicing variants of a single gene were annotated, the longest protein was chosen. A unique protein identifier was assigned to each protein, consisting of four letters and six numbers. The letters combine the first letter of the genus name with the first three letters of the species name. The versions and sources of the selected proteomes can be found in Table EV1.

To define phylogenetic profiles for all human proteins, we determined automatic orthologous groups (OG) across the database using

information from PANTHER 9.0 (Mi *et al*, 2016). PANTHER 9.0 contains 85 genomes within total of 1,136,213 genes. Of these genes, 759,627 genes are in PANTHER families with phylogenetic trees, multiple sequence alignments, and HMM profiles. In total, there are 7,180 PANTHER families and 52,768 subfamilies. Families are groups of evolutionary-related proteins and subfamilies are related proteins that are likely to have the same function. The division into subfamilies is done manually, by biological experts. Every subfamily of PANTHER is an OG at some taxonomic level in the tree of life. We used "hmmscan" tool from the HMMER package (Potter *et al*, 2018) (HMMER 3.1b1) to find for each protein sequence in our database, the best matching profile of a main family or subfamily in PANTHER9.0. The phylogenetic profile of panther main or subfamily was subsequently defined by utilizing the hierarchical nature of the panther classification. Specifically, the phylogenetic profile of a main or subfamily also includes all members of daughter families (and if relevant their daughter families, etc.). Note that due to the automatic nature of orthology definition and the draft quality of a few genomes, phylogenetic profiles of the human proteins are not as accurate as those defined by manual analysis (van Hooff *et al*, 2019).

To determine the phylogenetic profile similarity, Pearson's correlation (https://en.wikipedia.org/wiki/Phi_coefficient) was computed between the phylogenetic profile of the CENATAC panther (PTHR31198) and the phylogenetic profile of all other panther sub- and main families using in-house scripts. To detect functional patterns in orthologous groups with similar phylogenetic profiles (correlation > 0.5), a GO enrichment analysis was performed (Ashburner *et al*, 2000; Carbon *et al*, 2019; Mi *et al*, 2019). GO cellular component overrepresentation (GO Ontology database: released 2020-01-03) was computed using PANTHER (test release 2019-07-11) with the human reference genome gene set as background. Statistical significance of overrepresented GO terms was computed using Fisher's exact test with FDR correction.

### Nuclear extract and GFP pull-down and mass spectrometry

Nuclear extract of wild-type and HeLa$^{EGFP\text{-}CENATAC}$ cells was prepared as described earlier (Kloet *et al*, 2016). In short, cells were harvested by trypsinization and resuspended in cold hypotonic buffer (10 mM HEPES KOH pH 7.9, 1.5 mM MgCl$_2$, 10 mM KCl). Afterward, the cell pellet was homogenized using a Douncer with type B pestle (tight) to lyse the cell membrane. After centrifuging, the nuclei were washed with cold PBS and resuspended in cold buffer for lysis (420 mM NaCl, 20 mM HEPES KOH pH 7.9, 20% v/v glycerol, 2 mM MgCl$_2$, 0.2 mM EDTA) followed by rotation, centrifugation, and collection of the nuclear extract. 450 μl of nuclear extract was used for each GFP pull-down using 15 μl slurry of GFP-Trap agarose beads (Chromotek), performed in triplicate. GFP pull-downs were done as described earlier (Smits *et al*, 2013), without the addition of EtBr during the incubation, and with an adapted buffer C (150 mM NaCl, 20 mM HEPES KOH pH 7.9, 20 % v/v glycerol, 2 mM MgCl$_2$, 0.2 mM EDTA, complete protease inhibitors w/o EDTA, 0.5 mM DTT) for the incubation (+0.1% NP-40) and washes (+0.5% NP-40). Samples were digested using on-bead digestion with trypsin overnight (Hubner & Mann, 2011). The tryptic peptides were acidified with TFA and purified on C18 StageTips (Rappsilber *et al*, 2007).

After elution from the C18 StageTips, tryptic peptides were separated on an Easy-nLC 1000 (Thermo Scientific), connected online to a Q Exactive HF-X Hybrid Quadrupole-Orbitrap Mass Spectrometer (Thermo Scientific), using an acetonitrile gradient of 7–30% for 48 min followed by washes of 50–90% acetonitrile, for 60 min of total data collection. Full scans were measured with a resolution of 120,000, and the top twenty most intense precursor ions were selected for fragmentation with a resolution of 15,000 and dynamic exclusion set at 30 s. Peptides were searched against the UniProt human proteome (downloaded June 2017) using MaxQuant (Cox & Mann, 2008) (version 1.6.0.1) with default settings, and iBAQ, LFQ, and match-between-runs enabled. Data analysis was done using Perseus (version 1.5.5.3), and the volcano plot and stoichiometry calculations were done as described earlier (Smits *et al*, 2013) using in-house-made scripts for R (version 3.6.1).

### Nuclear extract preparations for Northern blots

Nuclear extract from HeLa S3 suspension cells was prepared according to the protocol described by Dignam *et al* (1983) using buffer D containing 50 mM KCL in the final dialysis step.

### Immunoprecipitation and Northern blots

100 μl nuclear extract diluted in lysis buffer to a final volume of 200 μl was incubated with 2 μg of anti-CCDC84 antibody (SIGMA-HPA071715) overnight in the cold room with end-to-end rotation. The following day capture of antibody–antigen complexes was done using 50 μl of resuspended Protein G Dynabeads prepared according to manufacturer's instructions and incubated with the nuclear extract antibody samples for 2 h at 4°C. Beads were then washed four times with lysis buffer lacking protease and RNase inhibitors. RNA was eluted by proteinase K treatment, extracted once with phenol:chloroform:isoamyl alcohol (25:24:1; pH 4.8) followed by ethanol precipitation. RNA was dissolved in H$_2$O or 0.1X TE buffer.

Total volumes of 2 μl (input) and 5 μl (IP) RNA samples were separated on a 6% polyacrylamide–urea gel and analyzed by Northern blotting essentially as described by Tarn and Steitz (1996). Individual snRNAs were detected using $^{32}$P 5′-end-labeled DNA or LNA oligonucleotides complementary to individual snRNAs. Northern blots were exposed to image plates and visualized using Typhoon FLA-9400 Scanner (GE Healthcare, USA) at 50-micron resolution. The data were quantified using AIDA Software (Raytest, Germany).

### Glycerol gradient and ultracentrifugation

HeLa S3 nuclear extracts were preincubated for 0–20 min at +30°C in a buffer containing 13 mM HEPES (pH 7.9), 2.4 mM MgCl$_2$, 20 mM creatine phosphate, 2 mM DTT, 40 mM KCl, and 0.5 mM ATP. Aggregates were subsequently removed by a brief centrifugation (20,000 *g*, 1 min, +4°C), and the supernatant was subsequently ultracentrifuged on a linear 10–30% glycerol gradient (20 mM HEPES, pH 7.9; 40 mM KCl, 2 mM DTT, 2.4 mM MgCl$_2$) for 18 h at 29,000 rpm, +4°C, Sorvall TH641 rotor (RCF(max) = 143,915.6 *g*). Following ultracentrifugation, the samples were fractionated. 20% of each fraction was deproteinized and used for RNA isolation and Northern blotting and the remaining 80% was subjected to TCA precipitation, separated on a 10% SDS–PAGE, and analyzed by

Western blots. Each blot was probed for CENATAC (CCDC84-HPA071715; Sigma-Aldrich–Merck), PRPF4 (#HPA0221794, Sigma-Aldrich–Merck).

### RT–PCRs

For Figs 4A and 4D, and Appendix Fig S10: Total cellular RNA was extracted using the RNeasy Kit Protocol (Qiagen) and treated with DNase I amplification grade (Invitrogen) to remove potential genomic DNA contamination. cDNA synthesis was carried out using SuperScript™ II RT (Thermo Fisher Scientific) and Oligo(dT)18 primers. PCRs were performed with Phusion High-Fidelity DNA Polymerase (Thermo Fisher Scientific) with the following cycling conditions: initial denaturation (98°C for 60 s), followed by 28–30 cycles of denaturing (98°C for 10 s), annealing (gene-specific temp. for 30 s), extension (72°C for 15–20 s), and a final extension (72°C for 1 min 30 s). PCR primers and relevant annealing temperatures are listed in Table EV3. PCR products were analyzed on 2% agarose gel run using 1X TBE buffer. For Figs 4B and 5E, total RNA isolated was isolated from HeLa cells or patient/control subject lymphoblasts using TRIzol extraction followed by an additional acidic phenol (pH 5.0) extraction. 1 μg of RNA was converted to cDNA using maxima H minus reverse transcriptase (Thermo Fisher) according to the manufacturer's protocol. PCRs were performed essentially as described above, and gene-specific primers and annealing temperatures are listed in Table EV3.

### RNA isolation and high-throughput sequencing

Total RNA isolated was isolated from HeLa$^{EGFP-AID-CENATAC}$ cells treated with siGAPDH (Dharmacon, D-001830-01-05) for 48 h or with siCENATAC (CCDC84, Dharmacon, J-027240-07) and 1 mM 3-indoleacetic acid (IAA) for 24 or 48 h, or unedited HeLa parental cells treated with siGAPDH for 48 h, or patient/control subject lymphoblasts using TRIzol extraction followed by an additional acidic phenol (pH 5.0) extraction. RNAseq libraries were constructed using Illumina TruSeq Stranded Total RNA Kit (Illumina) Human Ribo-Zero rRNA Depletion Kit (Illumina). Paired-end 150 + 150 bp sequencing was done with Illumina NextSeq 500/550 High Output Kit v2.5 for HeLa samples and with Illumina NovaSeq 6000 using partial S4 flow cell lane for patient samples.

### Mapping the reads to the genome

The STAR aligner (Dobin *et al*, 2013) was used for mapping the paired sequence reads to the genome (hg38/GRCh38). Transcript annotations were obtained from GENCODE (v29). The length of genomic sequence flanking the annotated junctions (sjdbOverhang parameter) was set to 161. The Illumina adapter sequences AGATCGGAAGAGCACACGTCTGAACTCCAGTCAC and AGATCGG AAGAGCGTCGTGTAGGGAAAGAGTGTAGATCTCGGTGGTCGCCGT ATCATT were, respectively, clipped from the 3′ of the first and the second pairs in the read libraries (using clip3pAdapterSeq parameter).

### Differential alternative splicing analysis

Differential AS analysis was done using Whippet (v0.11) (Sterne-Weiler *et al*, 2018). Both merged aligned reads (bam files) and AS

event annotations from GENCODE (v29) were used to build the index reference for AS events. To detect the significantly differential events, probability cutoff of Pr > 0.9 and percentage spliced in deviation cutoff of $|\Delta\Psi| > 0.1$ were used.

### Differential intron retention analysis

For a comprehensive and sensitive IR analysis, the IntEREst R/Bioconductor package was used (Oghabian *et al*, 2018). After reading binary alignment (.bam) files, IntEREst detects introns with significantly higher and lower number of mapped reads relative to the number of reads that span the introns. The DESeq2-based function of IntEREst, i.e., deseqInterest(), was used for the differential IR analysis. The Benjamini–Hochberg method was used for adjusting the *P*-values, and a cutoff of $P_{adj} < 0.05$ was applied to extract the significantly differential IRs. The reference table was built from the NCBI RefSeq transcription annotations based on hg38/GRCh38 genome assembly.

### Annotating minor introns

We used IntEREst R/Bioconductor package to annotate the minor (U12-type) introns as described previously (Oghabian *et al*, 2018) using threshold values of 0.07 and 0.14 for 5′ss and BPS scores, respectively. BPS was identified by scanning intronic region from position −40 to position −3 upstream of the 3′ss, and the highest scoring sequence was selected as the BPS. This list was manually appended with additional introns that did not fulfill our annotation criteria (typically because of poor BPS), but have been previously identified as minor introns (Chang *et al*, 2007).

### P120 minigene cloning, transfection, and analysis of RNA

The double 5′ss constructs were created by insertion mutagenesis PCR using the P120 minigene (Hall & Padgett, 1996) as a template, and further modifications of 5′ splice sites were made by PCR using mutagenic primers (for a list of primers used see Table EV3). The 3′ss was modified to accommodate for GT-subtype splicing by insertion of a CAG trinucleotide sequence through insertion mutagenesis PCR. All mutations were confirmed by DNA sequencing. Chinese hamster ovary cells were transfected with the double 5′ss constructs (1,600 ng per well of a 12-well plate) using Lipofectamine 2000 (Thermo Fisher Scientific), and after 24 h, total RNA was isolated using TRIzol reagent (Thermo Fisher Scientific). Following DNase treatment, a pCB6 vector-specific oligonucleotide (ACAGGGATGC CA) was used for reverse transcription of the RNA with RevertAid (Thermo Fisher Scientific). RT–PCR was performed with primers binding exon 6 (GGATGAGGAACCATTTGTGC) and exon 7 (AGAACGAGACCGCCCTTC), and the resulting PCR products were analyzed on a 3% MetaPhor™ (Lonza) agarose gel. The gel was imaged using Fuji LAS-3000 CCD Camera, and the band intensities were quantified using AIDA Software (Raytest, Germany). Identities of the PCR products were confirmed by DNA sequencing.

## Data availability

The authors declare that the data supporting the findings of this study are available within the paper and its supplementary

information. The ICR1000 UK exome series data are available at the European Genome-Phenome Archive (EGA), Reference Number EGAS00001000971 (https://ega-archive.org/studies/EGAS00001000 971). Exome data for individual patients cannot be made publicly available for reasons of patient confidentiality. Qualified researchers may apply for access to these data, pending institutional review board approval.

HeLa[EGFP-AID-CENATAC] RNAseq data were deposited in Gene Expression Omnibus GSE143392 (https://www.ncbi.nlm.nih.gov/geo/query/acc.cgi?acc=GSE143392). RNAseq data from the patient and control subject cannot be made publicly available for reasons of patient confidentiality. Qualified researchers may apply for access to these data, pending institutional review board approval.

Protein interaction AP-MS data were deposited in PRIDE PXD024682 (https://www.ebi.ac.uk/pride/archive/projects/PXD024 682).

Expanded View for this article is available online.

## Acknowledgements

We thank the patient family members for their participation in this study. We thank Anna Zachariou for assistance with recruitment, Emma Ramsay for performing the exome sequencing, and Elise Ruark for discussions about the analyses. We thank the Kops, Frilander, Snel, and Rahman laboratories for discussions and comments on the manuscript. We thank Andrew Holland for reagents. The Kops and Vermeulen labs are part of the Oncode Institute, which is partly funded by KWF Kankerbestrijding (DCS). This study was further funded by the Dutch Research Council (NWO) (OCENW.KLEIN.182), the Cancer Genomics Center (CGC.nl), the Wellcome Trust (100210/Z/12/Z) to NR, Sigrid Jusélius Foundation (MF), Jane and Aatos Erkko Foundation (MF), Academy of Finland grant 1308657 (MF), and a Postdoctoral Research Fellowship by the Herchel Smith Fund at the University of Cambridge (ET).

## Author contributions

BW, GJPLK, MJF, NR, and ECT conceptualized the data. BW, AO, MVA, SH, JEH, SY, ECT, LV, MPAB, ECHU, JV, LER, and PP investigated the data. BW, AO, MVA, SH, JEH, SY, ECT, LV, MPAB, ECHU, JV, LER, and PP involved in formal analysis. BW, AO, MVA, SH, JEH, SY, ECT, LV, MPAB, JV, PP, and MJF designed methodology. BW, AO, MVA, SH, JEH, SY, ECT, LV, MPAB, ECHU, JV, LER, and PP validated the data. BW, AO, MVA, SH, ECT, LV, JV, and LER visualized the data. BW, AO, MVA, SH, SY, LV, JV, and LER curated the data. AO and JEH provided software. BW, GJPLK, MJF, MVA, and NR wrote the original manuscript and prepared draft. BW, GJPLK, MJF, BS, MV, NR, BI, AO, MVA, SH, JEH, SY, ECT, LV, ECHU, JV, and LER wrote, reviewed, and edited the manuscript. GJPLK, MJF, BS, MV, NR, and BI administered the project. GJPLK, MJF, BS, MV, NR, and BI supervised the data. GJPLK, MJF, BS, MV, NR, and BI provided resources. GJPLK, MJF, BS, MV, NR, and BI acquired funding.

## Conflict of interest

Nazneen Rahman is a non-executive director of AstraZeneca. The other authors declare no competing interests.

## Materials and correspondence

Further information and requests for resources and reagents should be directed to and will be fulfilled by the lead contact, Geert Kops (g.kops@hubrecht.eu).

## Note

[1]CENATAC was named after AT-AC introns, which make up 84% of U12 A-type introns

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
