## [Review Process File · The EMBO Journal]

Chromosomal instability by mutations in the novel minor spliceosome component *CENATAC*

Bas de Wolf, Ali Oghabian, Maureen Akinyi, Sandra Hanks, Eelco Tromer, Jolien van Hooff, Lisa van Voorthuijsen, Laura van Rooijen, Jens Verbeeren, Esther Uijttewaal, Marijke Baltissen, Shawn Yost, Philippe Piloquet, Michiel Vermeulen, Berend Snel, Bertrand Isidor, Nazneen Rahman, Mikko Frilander, and Geert Kops

DOI: [10.15252/embj.2020106536](https://doi.org/10.15252/embj.2020106536)

Corresponding author(s): Geert Kops (g.kops@hubrecht.eu)

Review Timeline:

Submission Date:	14th Aug 20
Editorial Decision:	21st Sep 20
Revision Received:	26th Feb 21
Editorial Decision:	25th Mar 21
Revision Received:	16th Apr 21
Accepted:	19th Apr 21

Editor: Hartmut Vodermaier

Transaction Report:

Thank you for submitting your study on minor spliceosome links to chromosomal instability, and please excuse the fact that it has taken somewhat longer than usually to get back to you with a response. I have now received the reports from two splicing experts (refs 1 and 3) and two referees familiar with chromosome segregation/aneuploidy (refs 2 and 4), whose comments are copied below. Given that all referees find the study well done and the results potentially interesting, we would be happy to consider a revised manuscript further for EMBO Journal publication. As you will see, many points raised by the referees are quite specific and/or affecting aspects of presentation/discussion, but there are also a few more substantial concerns that I feel would be important to follow up on. In particular, both referees 2 and 4 ask for key key functional experiments to be validated in at least one additional cell type/cell line. Furthermore, these two reviewers are also concerned that the current data do not establish how splicing alterations in CENATAC mutants cause the MVA phenotype - I appreciate that this may be difficult to fully dissect, but at least some further validation that CENATAC mutations are causal to the CIN phenotype would be helpful, possibly via rescue experiments such as those suggested in ref 4 pt 6. On the other hand, while further discussions of the possible reasons for the discrepancies with Wang et al would be beneficial for answering referee 2's point 1, I do not see it warranted to conduct additional, detailed follow-up work solely for this purpose.

Referee #1:

In this manuscript, Wolf et al., have identified biallelic mutations in the CCDC84 (CENATAC) gene by

exome sequencing on patients with Mosaic variegated aneuploidy (MVA). These mutations create new splice sites leading to frameshifts and production of CCDC84 truncations. Knockdown of this protein in HeLa cells led to chromosome congression defects, and while this defect could be rescued by expression of the wild-type protein, the MVA mutants exacerbated it. Their co-evolution analysis as well as mass spectrometry analysis of proteins co-immunoprecipitated with CENATAC identified components of the spliceosome. Interestingly, CENATAC co-immunoprecipitation experiments revealed its association with the U12-type minor spliceosomal U4atac/U6atac/U5 tri-snRNPs. Depletion of CENATAC by siRNAs or expression of the MVA mutant variants affects splicing of selected minor introns in cells, which correlated with the extent of the mitotic phenotype. Consistent with these results, their transcriptomic analyses of CENATAC-depleted cells as well as MVA patient cells (containing CENATAC mutants) showed significant retention of a subclass of minor introns (with AT-AN dinucleotide splice sites, which the authors define as A-type minor introns). While a number of mitotic regulators were among affected transcripts, these did not contain any of the known MVA-related genes. Moreover, utilizing a minigene construct with competing A- or G- type splice sites, they demonstrate that A-type minor introns are spliced less efficiently and thus CENATAC facilitates splicing of this subtype of weak minor introns. Taken together, this work sheds light on the function of a novel minor tri-snRNP component whose mutations lead to MVA clinical phenotype. This is an interesting story, the experiments are carefully conducted, and the manuscript is well written. I strongly support publication of the study in its current form, with only minor revisions.

1. The authors conclude that a subset of minor introns with reduced intrinsic splicing activity is dependent on CENATAC activity and that it selectively promotes splicing of A-type minor introns. How the authors envisage the selectivity for 5'ss adenosine by CENATAC to occur? Could CENATAC simply act as a general enhancer, which facilitates splicing of less efficiently spliced minor introns, rather than a specificity factor for A-type introns? If the former is more likely, I recommend to remove "specificity" from the title as well as the abstract.
2. In figure 4B, it would be important to include a panel which shows U2-type splicing experiments as a control for the splicing of U12-type introns.
3. It is not clear from figure 1C why the truncation mutations lead to double bands in the western blots. Perhaps one band is the phosphorylated form, but the authors should comment on this point in the figure legend.
4. There are typos in figure legends 1E: "...in cells treated as in E).", S15 : "Top 20 categories are shows", Fig.6A: "...unchanged or retained after or ..."
5. In figure 3F, it would be helpful to indicate the gradient fraction numbers on the gel. Please mention also the type of gradient used in this experiment.
6. Please specify in the figure legends, which type of cells have been used in each experiment, as in some cases HeLa and in other cases patient lymphoblast cells have been used.

Referee #2:

In this manuscript, de Wolf et al. report that germline mutations in CENATAC cause chromosomal instability (CIN) in patients with Mosaic variegated aneuploidy (MVA). They first performed whole exome sequencing and identified biallelic truncating mutations in CENATAC in two affected

siblings. These mutations mildly increased CIN in affected lymphoblasts, and genetic depletion of CENATAC led to chromosome congression defects and mitotic arrest in HeLa cells. Using a co-evolution analysis, the authors identified CENATAC as a novel subunit of the minor (U12) spliceosome, and this was confirmed by mass spectrometry and by the retention of minor introns in several genes following CENATAC depletion. CENATAC depletion selectively inhibited the splicing of the A-type minor introns. CENATAC mutant patient cells showed high levels of A-type intron retention affecting a small set of genes that include genes associated with cell cycle and chromosome segregation. The authors therefore suggest that the MVA molecular phenotype of these MVA patient can be explained by the aberrant splicing of these genes.

Overall, this is a very interesting study that reveals an intriguing link between the minor spliceosome and aneuploidy in the context of a rare genetic disorder. The experiments are carried out carefully, and the manuscript is well written. The weaker aspect of the paper is that it doesn't really uncover how splicing aberrations lead to the molecular phenotypes of MVA. In particular, the study fails to establish a causal link between the splicing effect and the CIN induced by CENATAC depletion/mutation. This major limitation notwithstanding, the study will be high interest for the field, provided that it addresses a few key concerns, expands some of the analyses, and clearly acknowledges what remains unknown.

Major comments:

1) As the authors note, CENATAC was reported to localize to centrosomes and suppress centriole over-duplication and spindle multipolarity (ref 38; PMID: 31722219), which could potentially explain mitotic aberrations following CENATAC depletion. Interestingly, however, the current study did not identify centriole over-duplication in CENATAC-depleted cells. Moreover, whereas the previous study knocked out CENATAC in HeLa cells, the authors were not able to reproduce this KO in the current work. The discrepancies between the studies seem to be very important to the question at hand - how does CENATAC depletion promotes aneuploidy? While I appreciate that without obtaining the knockout cells this issue would be difficult to resolve, it would be important to address these discrepancies more thoroughly. Specifically, did the authors try to KO CENATAC the same way the same way reported by Wang et al? What could underlie the discrepancy, given that the same cell line was used? Importantly, the authors were able to achieve a very strong depletion of the protein by combining siRNAs and a degron system (Fig. S4); why is it then that they don't see the previously reported centriole over-duplication? The authors make the general suggestion that the differences may be related to how prolonged the CENATAC perturbation is - can this be tested in a time-course experiment? In its current form, it is very difficult to reconcile the manuscript with the recent published work of Wang et al.

2) "The extent of the splicing defect strongly correlated with the extent of the mitotic phenotype for all mutations (Fig. 4C), indicating that it is likely that the chromosome congression phenotype is a secondary effect of impaired minor spliceosome function." This correlation is very strong, but it doesn't really show that the mitotic aberrations are a consequence of the impaired minor spliceosome function (as both might be independently correlated with the degree of CENATAC inhibition in the cells). This statement should be toned down. To establish a stronger link between the minor spliceosome and CIN, can the authors knock down ZRSR2 (or other components of the minor spliceosome) and test whether this would lead to the same chromosome congression defects and aneuploidy observed in CENATAC-depleted cells and in MVA patients? (A good control would be to knock down components of the major spliceosome and show that the effect on CIN is specific to the minor spliceosome). This could provide an important support for the author's contention that aberrant splicing underlies the chromosomal phenotypes seen in CENATAC-

mutant MVA cells.

3) The functional annotation analysis of the genes affected by CENATAC depletion is rather limited. The authors state that "CENATAC mutant patient cells show high levels of A-type intron retention in only a small set of genes including ones involved in chromosome segregation." However, not much data are presented to support this statement. Rescue experiments with few single genes failed, but could the general analysis be expanded in order to identify genes of potential importance? For example, can GSEA be performed on the list of genes shown in Fig. 5d? Of note, in the Metascape analysis shown in Fig. S15 no functional annotations are directly related to chromosome congression or segregation, which seems a bit at odds with the authors' assertion that the aberrant splicing directly leads to the CIN phenotype.

4) All the functional experiments, including CENATAC depletion, rescue assays, mitotic analyses, proteomics and RNAseq are based on a single cell line (HeLa). For key experiments, at least one additional cell line is required in order to demonstrate the generalizability of the results. This seems to be particularly important for the RNAseq analysis, as both expression and splicing are expected to be strongly affected by the basal expression/splicing of the cell line. Adding one or two additional cell lines may make it easier to home in on important affected genes.

5) Related to the previous point, what is the role of tissue specificity in intron retention and minor spliceosome function? Is splicing efficiency different between tissue types, and could this be related to the tissue-specific phenotypes of the disease? An analysis of this question (perhaps with available RNAseq data from multiple tissues?), and a discussion of previous related literature, are in place.

6) The authors mention the importance of aneuploidy in cancer, but no cancer-related analysis is presented. Is CENATAC ever mutated in cancer? What is known about the activity of the minor spliceosome in tumors? Is there any association between minor spliceosome function/expression and high degree of CIN/aneuploidy? Do pediatric tumors that characterize MVA patients exhibit any splicing defects? There are available RNAseq data sets (from TCGA, for example) that can be used to address these questions.

7) As mentioned above, the claims about the potential link between the splicing effect and CIN effect of CENATAC perturbation should be presented in a more careful, speculative way, and a more thorough discussion of the various possibilities should be added.

Referee #3:

In this manuscript, the authors identify biallelic mutations in patients with a chromosomal aneuploidy disorder in the *CCDC84* gene (which they re-name CENATAC). They then validate the chromosomal effect in engineered HeLa cells and provide evidence that this protein has an essential function in the splicing of U12-type introns. In particular, they find that poorly spliced U12-type introns are most affected by loss of CENATAC function. These introns tend to be of the AT-AN subclass although they do find that some introns of the GT-AN subclass are also affected. Nevertheless, they provide significant data suggesting that the CENATAC protein has a function in minor class intron splicing.

Overall, this is a well constructed paper that makes a compelling case for its conclusions. There are

a few comments listed below. There are also a few typographical errors that should be fixed by spell-checking. Overall, though, the writing is fairly smooth.

Points to consider:

1. Early in the paper, there seems to be an effort to directly tie the function of CENATAC to chromosomal segregation but then the argument seems to shift to the idea that this is an indirect effect of the mis-splicing. For example, page 8 top paragraph last sentence, in contrast to page 10 top paragraph last sentence. I would suggest that the comments such as on page 8 be modified to allow for indirect effects since that's what they conclude with.
2. While they define the terms for the intron types, there are places where the terms are mixed in a confusing way. I think they should choose either major/minor or U2-type/U12-type and try to stick with it for the most part.

Minor correction:

The number for the estimated chance of mutations on page 6 line 6 is not valid.

Referee #4:

Review of the manuscript Chromosomal instability by mutations in a novel specificity factor of the minor spliceosome by de Wolf et al

Aneuploidy occurs through mitotic segregation errors and has severe physiological consequences as the leading cause of miscarriages and congenital defects, and as a hallmark of cancer. Only a few genes have been identified so far that are responsible for congenitally occurring aneuploidy. This is likely due to the fact that severe aneuploidy is embryonically lethal. The manuscript presents novel biallelic truncating mutations in CENATAC (CCDC84) that was identified via exome sequencing of patients with constitutional mosaic aneuploidy. They characterize CENATAC as a novel component of the minor (U12-dependent) spliceosome and show comprehensive data supporting the idea that CENATAC may promote splicing of a rare minor intron subtype. Depletion of CENATAC resulted in retention of AT-AN minor introns in ~100 genes that were enriched for nucleocytoplasmic transport and cell cycle regulators. The depletion also caused chromosome segregation errors. This is an interesting and well executed work that reveals a novel cause of congenital aneuploidy. The manuscript is well written, with nice figures supporting the claims. The authors elegantly characterize the splicing function of CENATAC, bringing thereby useful novel knowledge about minor spliceosome. However, the manuscript does not provide an explanation how exactly CENATAC affects the chromosomal instability and the link remains unclear. Several parts of the text should be changed or amended, and possibly some experiments need to be added before acceptance.

1. On page 6, the authors write that the expression of the maternal allele in patient 1 was responsible for the low expression of wild -type protein. How do they know this? Also, how was the situation with patient 2? In principal, patient 1 and 2 should have exactly the same phenotype. This should be shown and explained.
2. The separation of function of motif 3 and 4 versus motif 1 and 2 (page 10) is interesting. The authors may discuss this more.
3. The data from restoring experiments (page 12) should be shown. Did the authors try to overexpress all of them at the same time?
4. The analysis of mitotic errors in patient's cells reveals a very mild increase in mitotic errors (from 4 % in wild type to 8 % in mutant). Yet, the frequency of mitotic errors is very high in cell lines upon

depletion or expression of mutants. Why is that?

5. Why were the experiments performed in HeLa cells that are highly aneuploid and chromosomally unstable? At least some depletion should be performed in non-transformed cells to get data more comparable with the in vivo situation.

6. Can the authors rescue to mitotic defect in patients cell (e.g. in iPSC) by restoring the wt CENATAC expression?

7. Several isoforms and transcriptional variants of spindle and mitotic proteins have been previously identified (e.g. PMID: 20932937). The authors might compare the previously identified alternatively spliced variant with the targets of CENATAC.

8. Why it is necessary to rename the CCDC84 as CENATAC?

Rebuttal to reviewer comments:**Referee #1:**

[...] This is an interesting story, the experiments are carefully conducted, and the manuscript is well written. I strongly support publication of the study in its current form, with only minor revisions.

1. The authors conclude that a subset of minor introns with reduced intrinsic splicing activity is dependent on CENATAC activity and that it selectively promotes splicing of A-type minor introns. How the authors envisage the selectivity for 5'ss adenosine by CENATAC to occur? Could CENATAC simply act as a general enhancer, which facilitates splicing of less efficiently spliced minor introns, rather than a specificity factor for A-type introns? If the former is more likely, I recommend to remove "specificity" from the title as well as the abstract.

We currently do not know the exact function of CENATAC in the spliceosome or how it contributes to the preference for splicing A-type minor introns. This will require detailed mechanistic and structural studies as we have discussed in the Discussion section of the manuscript. We agree with the reviewer that it is possible that CENATAC could act as a general enhancing factor of splicing by the minor spliceosome. We have further clarified this in the Discussion and changed the title.

2. In figure 4B, it would be important to include a panel which shows U2-type splicing experiments as a control for the splicing of U12-type introns.

We have now added such data to panel B of figure 4.

3. It is not clear from figure 1C why the truncation mutations lead to double bands in the western blots. Perhaps one band is the phosphorylated form, but the authors should comment on this point in the figure legend.

We have changed the figure legend to better explain these bands.

4. There are typos in figure legends 1E: "...in cells treated as in E).", S15 : "Top 20 categories are shows", Fig.6A: "...unchanged or retained after or ..."

The errors have been corrected.

5. In figure 3F, it would be helpful to indicate the gradient fraction numbers on the gel. Please mention also the type of gradient used in this experiment.

This information has been added to the figure and legend.

6. Please specify in the figure legends, which type of cells have been used in each experiment, as in some cases HeLa and in other cases patient lymphoblast cells have been used.

This information has been added to the figure and legend.

Referee #2:

[.....] Overall, this is a very interesting study that reveals an intriguing link between the minor spliceosome and aneuploidy in the context of a rare genetic disorder. The experiments are carried out carefully, and the manuscript is well written. The weaker aspect of the paper is that it doesn't really uncover how splicing aberrations lead to the molecular phenotypes of MVA. In particular, the study fails to establish a causal link between the splicing effect and the CIN induced by CENATAC depletion/mutation. This major limitation notwithstanding, the study will be high interest for the field, provided that it addresses a few key concerns, expands some of the analyses, and clearly acknowledges what remains unknown.

Major comments:

1) As the authors note, CENATAC was reported to localize to centrosomes and suppress centriole over-duplication and spindle multipolarity (ref 38; PMID: 31722219), which could potentially explain mitotic aberrations following CENATAC depletion. Interestingly, however, the current study did not identify centriole over-duplication in CENATAC-depleted cells. Moreover, whereas the previous study knocked out CENATAC in HeLa cells, the authors were not able to reproduce this KO in the current work. The discrepancies between the studies seem to be very important to the question at hand - how does CENATAC depletion promote aneuploidy? While I appreciate that without obtaining the knockout cells this issue would be difficult to resolve, it would be important to address these discrepancies more thoroughly. Specifically, did the authors try to KO CENATAC the same way the same way reported by Wang et al? What could underlie the discrepancy, given that the same cell line was used? Importantly, the authors were able to achieve a very strong depletion of the protein by combining siRNAs and a degron system (Fig. S4); why is it then that they don't see the previously reported centriole over-duplication? The authors make the general suggestion that the differences may be related to how prolonged the CENATAC perturbation is - can this be tested in a time-course experiment? In its current form, it is very difficult to reconcile the manuscript with the recent published work of Wang et al.

We agree with the reviewer that this discrepancy is important to resolve. As the reviewer points out, it will require a substantial effort to do so. After many attempts we have been unable to obtain knock-out cells, which to us is not surprising given that *CCDC84* was found to be an essential gene in various studies [PMID 26472760, 26472758, 26627737]. In our degron/RNAi approach, we do see spindle pole problems appearing after prolonged mitotic delays, without evidence of centriole overduplication. Longer depletions in our hands lead to cell death, prohibiting testing of the reviewer's suggestion. Given our current attempts and the substantial effort needed to figure out why we don't see the phenotype of Wang et al., we hope the reviewer agrees that we can instead focus on the positive results, found consistently in patient cells and HeLa cells: that CENATAC is a minor spliceosome component. As described in the discussion, we acknowledge the possibility that spindle pole/centrosome defects additionally contribute to CIN in the patients.

2) "The extent of the splicing defect strongly correlated with the extent of the mitotic phenotype for all mutations (Fig. 4C), indicating that it is likely that the chromosome congression phenotype is a secondary effect of impaired minor spliceosome function." This correlation is very strong, but it doesn't really show that the mitotic aberrations are a consequence of the impaired minor spliceosome function (as both might be independently correlated with the degree of CENATAC inhibition in the cells). This statement should be toned down.

We have changed the sentence to "The extent of the splicing defect strongly correlated with the extent of the mitotic phenotype for all mutations (Fig. 4C), supporting the possibility that impaired minor spliceosome function and the chromosome congression phenotype are causally linked."

To establish a stronger link between the minor spliceosome and CIN, can the authors knock down ZRSR2 (or other components of the minor spliceosome) and test whether this would lead to the same chromosome congression defects and aneuploidy observed in CENATAC-depleted cells and in MVA patients? (A good control would be to knock down components of the major spliceosome and show that the effect on CIN is specific to the minor spliceosome). This could provide an important support for the author's contention

that aberrant splicing underlies the chromosomal phenotypes seen in CENATAC-mutant MVA cells.

We have now added data showing that ZRSR2 depletion by RNAi in HeLa and DLD-1 cells causes a chromosome congression defect similar to CENATAC depletion (Figs. 4D-E), providing independent support for a role of the minor spliceosome in chromosome segregation. We have not pursued depletion of the major spliceosome as control, because as major introns are present in virtually every gene, it is to be expected that general splicing defects will impact most if not all cellular processes, including chromosome segregation, as was shown previously (e.g. PMID 25257309, 25092791, 25257310).

3) The functional annotation analysis of the genes affected by CENATAC depletion is rather limited. The authors state that "CENATAC mutant patient cells show high levels of A-type intron retention in only a small set of genes including ones involved in chromosome segregation." However, not much data are presented to support this statement. Rescue experiments with few single genes failed, but could the general analysis be expanded in order to identify genes of potential importance? For example, can GSEA be performed on the list of genes shown in Fig. 5d? Of note, in the Metascape analysis shown in Fig. S15 no functional annotations are directly related to chromosome congression or segregation, which seems a bit at odds with the authors' assertion that the aberrant splicing directly leads to the CIN phenotype.

We have now included a more extended and more detailed analysis of the genes found to have high intron retention levels in order to more clearly show that this list is enriched for mitotic regulators (see Fig. S17). The supplementary table has also been reorganized to make it easier to identify the genes related to each GO category.

4) All the functional experiments, including CENATAC depletion, rescue assays, mitotic analyses, proteomics and RNAseq are based on a single cell line (HeLa). For key experiments, at least one additional cell line is required in order to demonstrate the generalizability of the results. This seems to be particularly important for the RNAseq analysis, as both expression and splicing are expected to be strongly affected by the basal expression/splicing of the cell line. Adding one or two additional cell lines may make it easier to home in on important affected genes.

We demonstrate CENATAC's splicing phenotype in two different cell lines: in HeLa cells (upon CENATAC depletion) and in patient lymphoblasts (CENATAC mutations). The splicing defects are highly concordant: in both lines it is predominantly the A-type minor introns that are affected, and virtually all affected genes in the lymphoblast are also found affected in the HeLa cell experiments as presented in Figure 5G. To strengthen our conclusions about the mitotic phenotype, in addition to showing chromosome congression defects upon depletion of ZRSR2, we now show that MVA-mutant CENATAC causes chromosome congression defects in DLD-1 cells also, identical to those upon CENATAC depletion by degron/RNAi or ZRSR2 RNAi in HeLa cells (Fig.S7, Fig. 4).

5) Related to the previous point, what is the role of tissue specificity in intron retention and minor spliceosome function? Is splicing efficiency different between tissue types, and could this be related to the tissue-specific phenotypes of the disease? An analysis of this question (perhaps with available RNAseq data from multiple tissues?), and a discussion of previous related literature, are in place.

We agree that this is a very interesting question. The tissue-specific phenotypes of the CENATAC-mutated patients resemble that of MVA in general and may thus be unrelated to possible tissue-specific effects of the minor spliceosome. However, we unfortunately do not have patient samples or cell lines derived from other tissues to examine this. Furthermore, because of the very limited data available on minor spliceosome tissue-specificity, we are unable to properly address this at this moment.

6) The authors mention the importance of aneuploidy in cancer, but no cancer-related analysis is presented. Is CENATAC ever mutated in cancer? What is known about the activity of the minor spliceosome in tumors? Is there any association between minor spliceosome function/expression and high degree of CIN/aneuploidy? Do pediatric tumors that characterize MVA patients exhibit any splicing defects? There are available RNAseq data sets (from TCGA, for example) that can be used to address these questions.

Cancer in the MVA patients is mostly associated with the *BUB1B* and *TRIP13* gene mutations. No tumors were reported in *CEP57* or *CENATAC* mutant patients. This, in addition to the fact that *CENATAC* mutations cause more specific splicing defects than general minor spliceosome defects, leads us to consider it unlikely that a strong correlation between minor spliceosome defects and cancer exists. This is in line with current knowledge: the only example thus far of an association between minor spliceosome malfunction and cancer are *ZRSR2* mutations that lead to myelodysplastic syndrome which can develop into AML.

7) As mentioned above, the claims about the potential link between the splicing effect and CIN effect of *CENATAC* perturbation should be presented in a more careful, speculative way, and a more thorough discussion of the various possibilities should be added.

We have changed the text accordingly.

Referee #3:

[...] Overall, this is a well constructed paper that makes a compelling case for its conclusions. There are a few comments listed below. There are also a few typographical errors that should be fixed by spell-checking. Overall, though, the writing is fairly smooth.

Points to consider:

1. Early in the paper, there seems to be an effort to directly tie the function of *CENATAC* to chromosomal segregation but then the argument seems to shift to the idea that this is an indirect effect of the mis-splicing. For example, page 8 top paragraph last sentence, in contrast to page 10 top paragraph last sentence. I would suggest that the comments such as on page 8 be modified to allow for indirect effects since that's what they conclude with. We have changed the sentence on page 8 (now page 9) to "Taken together, these data show that *CENATAC* directly or indirectly promotes chromosome congression in mitosis (in a manner likely unrelated to its role in maintaining spindle bipolarity), and that MVA-mutant *CENATAC* is a defective variant".

2. While they define the terms for the intron types, there are places where the terms are mixed in a confusing way. I think they should choose either major/minor or U2-type/U12-type and try to stick with it for the most part.

We have changed the text to consistently refer to these introns as major/minor.

Minor correction:

The number for the estimated chance of mutations on page 6 line 6 is not valid.

We thank the reviewer for pointing this out, and have corrected it (now page 7).

Referee #4:

[...] This is an interesting and well executed work that reveals a novel cause of congenital aneuploidy. The manuscript is well written, with nice figures supporting the claims. The authors elegantly characterize the splicing function of *CENATAC*, bringing thereby useful novel knowledge about minor spliceosome. However, the manuscript does not provide an explanation how exactly *CENATAC* affects the chromosomal instability and the link

remains unclear. Several parts of the text should be changed or amended, and possibly some experiments need to be added before acceptance.

1. On page 6, the authors write that the expression of the maternal allele in patient 1 was responsible for the low expression of wild-type protein. How do they know this? Also, how was the situation with patient 2? In principal, patient 1 and 2 should have exactly the same phenotype. This should be shown and explained.

Our statement is based on the fact that only the maternal allele still contains the original splice site, which according to our RNAseq data is being used at low levels (see Fig. S2). Based on the sequence context we can distinguish the maternal and paternal alleles and their splice site usage from the RNAseq reads. The paternal mutation results in deletion of the original splice site, which means no wild-type protein can be expressed from it. Unfortunately we do not have samples or cell lines available of patient 2. It should indeed have the same phenotype as patient 1.

2. The separation of function of motif 3 and 4 versus motif 1 and 2 (page 10) is interesting. The authors may discuss this more.

We have now added a brief section on this in the discussion (page 17).

3. The data from restoring experiments (page 12) should be shown. Did the authors try to overexpress all of them at the same time?

We recognize that it would be a valuable experiment to over-express all splice targets at the same time, but unfortunately we have not been able to do this in such a way that we are able to validate their expression in single cells. However, since we were now able to show a strong mitotic congression defect upon the depletion of the minor spliceosome factor ZRSR2 (as suggested by referee 2, point 2), thereby providing independent support for a role of the minor spliceosome in chromosome segregation, we decided to remove the mention of these rescue experiments. This will allow us to address these experiments, and the above mentioned challenges, in a follow-up project.

4. The analysis of mitotic errors in patient's cells reveals a very mild increase in mitotic errors (from 4 % in wild type to 8 % in mutant). Yet, the frequency of mitotic errors is very high in cell lines upon depletion or expression of mutants. Why is that?

We think this is most likely due to the fact that whereas depletion of CENATAC in HeLa cells is very efficient (see Fig S4B), a substantial amount of wild-type protein is being expressed in the patient cells (see Fig 1C). Consistently, our RNAseq analysis also revealed that compared to even 24 hr depletion in HeLa cells, the MVA patient cells show lower level of splicing defect both at the average intron retention levels (Fig 5F) and at the number of genes affected (Fig 5G). The even higher penetrance of the phenotype when overexpressing the MVA mutants in HeLa cells further underscores this: the small amount of residual endogenous CENATAC upon depletion in HeLa cells is further inactivated by a dominant-negative effect of overexpressed mutant CENATAC.

5. Why were the experiments performed in HeLa cells that are highly aneuploid and chromosomally unstable? At least some depletion should be performed in non-transformed cells to get data more comparable with the in vivo situation.

Although we have been unable to sufficiently deplete CENATAC in non-transformed cells, we have now added data on near-diploid human DLD-1 cells, where expression of MVA-mutant CENATAC causes the same chromosome congression and minor intron splicing phenotypes as in HeLa cells (Fig. S7 and S12)

6. Can the authors rescue to mitotic defect in patients cell (e.g. in iPSC) by restoring the wt CENATAC expression?

We agree this would be interesting, but since we did not observe a robust mitotic defect to rescue in the patient lymphoblasts, we have not attempted such a rescue. It may indeed be possible that other cell types exhibit a more severe mitotic defect in culture, however, given the low level of aneuploidy observed in the patients (also compared to other MVA patients), we think this is a low probability. The effort of creating iPSC cells and subsequently restoring CENATAC expression is thus unlikely to result in more clear outcomes. Instead, we have created HeLa and DLD-1 cells expressing the MVA mutants, which have clear mitotic defects. That, in combination with aneuploidy in CENATAC mutant patients, makes a strong case that CENATAC mutations underlie the aneuploidy/CIN phenotype. We therefore hope the reviewer agrees that pursuing rescues in patients is not worth the effort at the moment.

7. Several isoforms and transcriptional variants of spindle and mitotic proteins have been previously identified (e.g. PMID: 20932937). The authors might compare the previously identified alternatively spliced variant with the targets of CENATAC.

Minor introns as such are not generally alternatively spliced, unlike the major introns. The alternative splice site activation relates to cryptic splice site activation near the defectively spliced U12-type introns. These are not the physiological targets of the alternative splicing regulation.

8. Why it is necessary to rename the CCDC84 as CENATAC?

We opted to do so in order to get away from a non-descript and hard-to-pronounce name. In deliberations with the authors of the Wang et al study on CCDC84 function in centriole duplication (the only other study to report on a function of the CCDC84 protein) and with a member of the HUGO gene nomenclature committee (HGNC), we agreed to the name CENATAC to indicate its molecular functions.

Thank you for submitting your revised manuscript for our editorial consideration. It has now been re-reviewed by three of the original referees (see comments below), who found all initially raised concerns satisfactorily addressed. After incorporation of the remaining editorial modifications listed below, we shall therefore be happy to accept the study for EMBO Journal publication.

Referee #1:

In the revised manuscript, the authors have corrected my points and conducted additional experiments and controls, and in my opinion, they have sufficiently addressed most of the points raised by the other referees.

Although, I noted that the Supp. figure numbers in the rebuttal letter were incorrect (e.g. Fig.S7 and S12 in the letter are actually S6 and S10 in the main text), but they appeared to be correctly referenced in the main text.

I think the revised manuscript is suitable for publication in EMBO J.

Referee #2:

The authors provide a few additional key experiments that improve the manuscript. In particular, the mitotic defects shown with the DLD1 cell line when a mutant CENATAC was introduced, and the ZRSR2 knockdown experiments, strengthen the link between minor intron retention and mitotic defects. While the molecular mechanism underlying this link remains obscure, the observation itself is quite interesting. I have no further comments.

Referee #4:

The authors addressed my comments sufficiently and I recommend the manuscript for publication.

Corresponding Author Name: Geert JPL Kops

Manuscript Number: 2020-106536